# STABLE DIFFUSION MODELS ARE SECRETLY GOOD AT VISUAL IN-CONTEXT LEARNING

## ABSTRACT

Large language models (LLM) in natural language processing (NLP) have demonstrated great potential for in-context learning (ICL) — the ability to leverage a few set of example prompts to adapt to various tasks without having to explicitly update model weights. ICL has recently been explored for the visual domain with promising early outcomes. These approaches involve specialized training and/or additional data which complicate the process and limit its generalizability. In this work, we show that off-the-shelf Stable Diffusion models can be re-purposed for visual in-context learning (V-ICL). Specifically, we formulate an in-place attention re-computation within the self-attention layers of the Stable Diffusion architecture that explicitly incorporates context between the query and example prompts. Without any additional fine-tuning, we show that this re-purposed Stable Diffusion model is able to adapt to six different tasks: foreground segmentation, single object detection, semantic segmentation, keypoint detection, edge detection, and colorization. For example, the proposed approach improves the mean intersection over union (mIoU) for the foreground segmentation task on Pascal-5i dataset by 8.9% and 3.2% over recent methods such as Visual Prompting and IMProv, respectively. Additionally, we show that the proposed method is able to effectively leverage multiple prompts through ensembling to infer the task better and further improve the performance across all tasks.

## 1 INTRODUCTION

*In-context learning (ICL)* refers to the paradigm where a large language model (LLM) or a foundation model leverages exemplar source-target pair(s), known as *prompts*, to infer the task and perform the inferred task on an input, known as *query*. ICL is an emergent property of LLMs and has been widely explored in the natural language processing (NLP) domain (Wei et al., 2022; Brown et al., 2020; Hao et al., 2022; Touvron et al., 2023). ICL facilitates a model to adapt to novel or out-of-domain tasks without the need for fine-tuning which not only eliminates task-specific training but also reduces dependency on task-specific annotated datasets. Recent research, such as Bar et al. (2022); Xu et al. (2023); Wang et al. (2023a;b;c); Liu et al. (2023); Wang et al. (2024), have made promising attempts to harness the potential of in-context learning for computer vision tasks. These works can be broadly divided into two categories based on the datasets that are used for training: (1) uncurated datasets (*e.g.* CVF (Bar et al., 2022), S2CV (Xu et al., 2023)) and (2) curated/annotated datasets (*e.g.* COCO (Lin et al., 2014), NYUDv2 (Silberman et al., 2012)).

The first category of approaches such as Visual Prompting (Bar et al., 2022) and Improv (Xu et al., 2023) enable the foundation model for in-context learning by training with inpainting loss on uncurated datasets (*e.g.* CVF, S2CV), which consists of images with grid-like structures of source-target pairs extracted from computer vision papers. The grid-like structure of input images along with the in-painting loss that is based on random masking allows the model to implicitly learn the relationships between the source and target images. During inference, when the model is presented with a grid-like canvas of the query image and the example prompt(s) (source-target pairs) along with the location of the output for the query image masked-out (see Fig. 1a), the model would inpaint the prediction.

The second category of approaches (Wang et al., 2023a;b;c; Liu et al., 2023; Wang et al., 2024) enable a foundation model for in-context learning by allowing the model to train on curated datasets

Figure 1: (a) Existing approaches like Visual Prompting (Bar et al., 2022) train a foundation model on uncurated datasets for the inpainting task. At inference, a grid of query and example prompt are provided as input to the model. (b) Visual Prompting (Bar et al., 2022) struggles to predict accurately even when the prompt is the same as the query. (c) The proposed approach, when presented with the same prompt as the query, is able to fully leverage the prompt to make accurate predictions.

(*e.g.* COCO, NYUDv2) related to the out-of-domain tasks that the model is expected to adapt to. While these methods do not employ task-specific loss functions, they still rely on task-representative datasets. For example, to enable the Painter (Wang et al., 2023a) model to adapt to open-vocabulary segmentation (*e.g.* FSS-1000 (Li et al., 2020)), the model is trained on a related dataset like the COCO segmentation dataset. In other words, as long as the model has encountered a task during training, it can perform that task on query images from unseen categories. However, this implies that the model needs to access large annotated datasets during training thereby undermining the key advantage of V-ICL. Furthermore, the generalization of such models is limited to tasks related to those seen during training. These methods[1] show promising early outcomes of vision foundation models on unseen tasks, however, achieving similar success as LLMs in NLP domain with off-the-shelf vision foundation models (*e.g.* Stable Diffusion (Rombach et al., 2022), ViT (Dosovitskiy et al., 2020)) remains an area yet to be fully explored.

Although these existing V-ICL approaches show promising outcomes, they involve the use of additional training steps and/or data from out-of-domain tasks to which the model is being used to adapt. Different from existing approaches, in this work, we focus on an approach that is more in-line with the ICL methods of the NLP community. Specifically, we address the question — *Can we use off-the-shelf foundation models for visual in-context learning without any additional training steps or data?* To this end, we propose the Stable Diffusion based visual in-context learning (SD-VICL) pipeline where we show that an off-the-shelf Stable Diffusion model can be re-purposed, without any fine-tuning, to adapt to several new out-of-domain tasks (see Fig. 2). We formulate an in-place attention re-computation within the self-attention layers of the Stable Diffusion architecture that explicitly incorporates the context between the query and example prompts. This formulation is based on our analysis of existing approaches like Visual Prompting (Bar et al., 2022) as shown in Fig. 1, where these models are unable to fully leverage the example prompt.

Furthermore, we note that existing approaches such as Bar et al. (2022); Xu et al. (2023) demonstrate the benefits of using multiple example prompts to boost the prediction abilities of the V-ICL foundation models. However, their approach of ensembling in the image space by creating a composite of multiple examples comes at the cost of reducing the resolution per prompt — limiting the potential of prompt ensembling. To address this issue, inspired by feature ensembling in SegGPT (Wang et al., 2023b), we perform prompt ensembling in the latent space for in-context learning. Note that the feature ensembling in SegGPT involves averaging of query features after each attention layer which results in uniform weighting of all prompts. In contrast, we perform implicitly-weighted prompt ensembling within each attention layer thereby enabling the model to better infer the information from multiple prompts.

We conduct extensive evaluations to demonstrate the effectiveness of the proposed approach. Specifically, we show that a pre-trained Stable Diffusion model has the ability to adapt to various tasks like foreground segmentation, single object detection, semantic segmentation, keypoint detection, edge detection, and colorization.

---

[1]For a more thorough discussion on related work, we refer the reader to the expanded discussion in Sec. B of the supplementary.

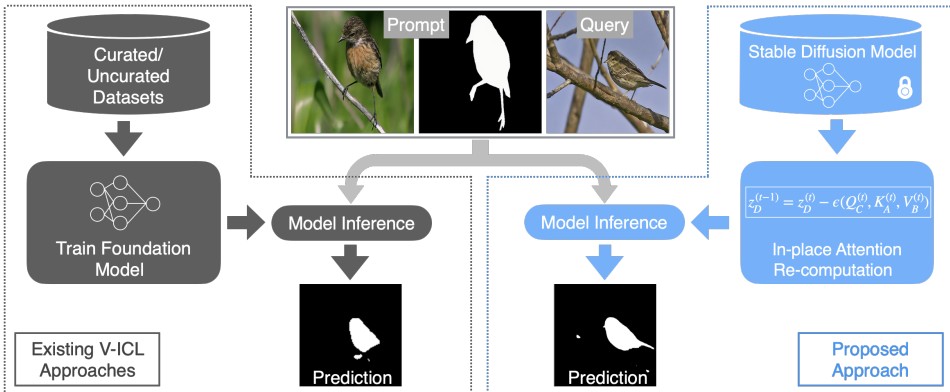

Figure 2: Existing V-ICL approaches rely on specialized training of foundation models on curated/uncurated data before using them for novel downstream tasks. The proposed approach re-purposes an off-the-shelf Stable Diffusion model without the need for training or data.

To summarize, the main contributions of this work are as follows,

- The first training-free method to enable visual in-context learning properties within a foundation model, setting a new direction for visual in-context learning research.
- A novel pipeline that explicitly incorporates the context between the query image and the prompts by introducing an in-place attention re-computation within the self-attention layers of an off-the-shelf Stable Diffusion model.
- Extensive evaluation of the proposed approach to demonstrate its ability to generalize to multiple out-of-domain tasks.
- We show that implicitly-weighted prompt ensembling enables effective use of multiple prompts for visual in-context learning.

## 2 METHOD

### 2.1 MOTIVATION AND PRELIMINARIES

In ICL, a foundation model is expected to infer the task based on a few source-target example prompts and predict the appropriate output for the query image. More importantly, as demonstrated in the NLP community, in-context learning is supposed to be an emergent property of a foundation model where the model does not require additional training. However, existing visual in-context learning approaches involve training and/or use of additional task-related data. Further, when we investigated V-ICL approaches based on uncurated datasets (*e.g.* Bar et al. (2022))[2] through multiple experiments, we observe the following, (1) the model fails to reconstruct the input prompt, despite it being unmasked in the input, as shown in the top-left image of the foundation model's output in Fig. 1a), (2) the model struggles in the presence of multiple objects in the image as shown in Fig. 1a) where it incorrectly predicts the *bird feeder* as the foreground, and (3) when provided with the query and its ground-truth as the example prompt, the model struggles to make correct predictions despite having access to the ground-truth for the query (see Fig. 1b). These observations indicate that the model is unable to fully leverage the prompts. We attribute this inability of the model to lack of appropriate context interpretation (1) between the source and target of the example prompt which is essential for task inference, and (2) between the query and the source of the example prompt which is essential for accurate predictions.

To alleviate these limitations, we propose a novel inference-only Stable Diffusion based visual in-context learning pipeline (SD-VICL), that unlike existing approaches, requires no additional training. Stable Diffusion (Rombach et al., 2022) is a latent diffusion model that generates high-quality images by iteratively refining random noise through a denoising process. The denoising process at a given time step $t$ employs a denoising U-Net which comprises of multiple self-attention layers

---

[2]We use approaches based on uncurated datasets as the basis for our analysis and formulations since they do not explicitly use images from out-of-domain tasks for training and are more closer to in-context learning in the NLP community as compared to the approaches that use curated datasets.

operating at resolutions $16 \times 16$, $32 \times 32$, and $64 \times 64$. At each self-attention layer ($l$), the input features of the intermediate noise latent, $\phi(z_t)$, are transformed to Query ($Q$), Key ($K$), and Value ($V$) vectors using linear layers. The $Q$ and $K$ vectors are used to compute the self-attention map using:

$$\alpha^{(t)} = \text{softmax}\left(\frac{Q^{(t)} \cdot K^{(t)^T}}{\sqrt{d}}\right), \tag{1}$$

which captures the correspondences within the image. The parameter $d$ denotes the feature dimension of the $Q$ vector. The self-attention map ($\alpha$) is used to update the intermediate feature map using the feature update, $\Delta\phi$, which is computed as,

$$\Delta\phi^{(t)} = \alpha^{(t)} \cdot V^{(t)}. \tag{2}$$

This updated set of features, along with the updates from the cross-attention layers are used to compute the intermediate noise prediction. In this paper, we focus only on the self-attention computations. Hence, for simplicity, we write the iterative denoising process as

$$z^{(t-1)} = z^{(t)} - \epsilon(Q^{(t)}, K^{(t)}, V^{(t)}), \tag{3}$$

where the predicted noise, $\epsilon$, is a function of the $Q$, $K$, and $V$ vectors of the self-attention layers.

## 2.2 RE-PURPOSING STABLE DIFFUSION FOR VISUAL IN-CONTEXT LEARNING (SD-VICL)

As discussed previously, for successful in-context learning, a foundation model needs to appropriately infer (1) *the task*: the relationship between the prompt image ($A$) and prompt ground-truth ($B$), and (2) *the context*: the relationship between the query image ($C$) and the prompt image ($A$). In an attempt to ensure that these relationships are effectively inferred, we formulate a novel attention computation in place of the traditional self-attention computation in the upsample layers of the denoising U-Net. This formulation is inspired by a few recent research (Cao et al., 2023; Alaluf et al., 2024; Tumanyan et al., 2023; Patashnik et al., 2023) that re-purpose the self-attention layers of Stable Diffusion for tasks like image editing and style transfer.

First, each of the input images: prompt image ($A$), prompt ground-truth ($B$), and query image ($C$) are inverted to the Stable Diffusion's noise space to obtain, $z_A^{(T)}$, $z_B^{(T)}$, and $z_C^{(T)}$ respectively, using an off-the-shelf inversion model (Huberman-Spiegelglas et al., 2024). Then, each of these images are iteratively denoised in parallel without any changes to the default denoising pipeline as follows:

$$z_p^{(t-1)} = z_p^{(t)} - \epsilon(Q_p^{(t)}, K_p^{(t)}, V_p^{(t)}), \qquad \text{where} \quad p \in \{A, B, C\}, t \in [T, 1], t \in \mathbb{Z}^+. \tag{4}$$

Since the output image is expected to be structurally similar to the query image for most vision tasks, we initialize the noise space in the prediction pipeline $D$ using the noise space of the query image (*i.e.*, $z_D^{(T)} = z_C^{(T)}$).

Different from Eq. (4), the query prediction is denoised using,

$$z_D^{(t-1)} = z_D^{(t)} - \epsilon(Q_C^{(t)}, K_A^{(t)}, V_B^{(t)}), \qquad \text{where} \quad t \in [T, 1], t \in \mathbb{Z}^+. \tag{5}$$

If we expand the feature update formulation for $D$ we get,

$$\Delta\phi_D^{(t)} = \alpha_D^{(t)} \cdot V_B^{(t)}, \tag{6}$$

$$\Delta\phi_D^{(t)} = \text{softmax}\left(\frac{Q_C^{(t)} \cdot K_A^{(t)^T}}{\tau \cdot \sqrt{d}}\right) \cdot V_B^{(t)}. \tag{7}$$

In contrast to Eqs. (1) and (2), the attention map is formulated using the Query vector of the query image ($C$) and the Key vector of the prompt image ($A$). The $Q_C$ vector comprises of the semantics of each spatial location in $C$, and the $K_A$ vector offers the context within $A$ that each query can attend to. Hence, each element in the attention map, *i.e.*, $\alpha_{D(i,j)}^t$, captures how the $i^{th}$ patch of the query image correlates with the $j^{th}$ patch of the prompt image. This formulation explicitly enforces the context between the query image and the prompt image. The additional temperature hyperparameter ($\tau$) (in Eq. (7)) for the softmax computation controls the sharpness of the correlation. By adjusting

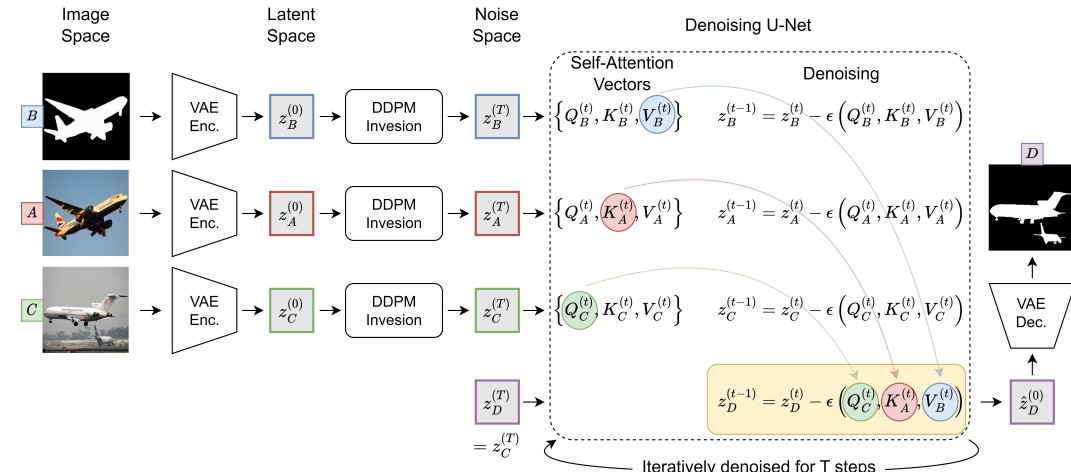

Figure 3: Latent representations of the query and prompt images from the Stable Diffusion model's VAE encoder are transformed to the noise space using (Huberman-Spiegelglas et al., 2024). Each noisy latent is iteratively denoised using the denoising U-Net. Prediction ($D$) path is initialized with the noise space of $C$, and at each denoising step, the computations within the self-attention layers are modified to enhance visual ICL capabilities. With the Query (from prompt image-$A$), the Key (from prompt ground-truth-$B$), and the Value (from query image-$C$) vectors, we reformulate the intermediate feature update to infuse context between the example prompts and the query image into the foundation model. This process is iteratively performed for $T$ denoising steps to obtain the denoised prediction latent, which is then fed to the VAE decoder to obtain the final prediction $D$.

$\tau$, we can modulate the model's focus on attention/correlation between $C$ and $A$. A lower $\tau$ sharpens the attention, emphasizing stronger correlation, while a higher $\tau$ smoothens the attention, allowing for a broader distribution of focus across multiple correlations. Furthermore, we use the Value vector from the prompt ground-truth ($B$) to compute $\Delta\phi_D^{(t)}$. This use of value vector from a different latent, in contrast to the standard implementation of cross-attention (Vaswani, 2017), where both Key and Value vectors typically come from the same set of intermediate latents, is to facilitate the prediction to be in the same domain as that of the prompt ground-truth $B$. This formulation, which is in-place of the computation within the standard self-attention layer of Stable Diffusion, facilitates the explicit infusion of the context between the query image and the prompt in addition to improved task inference. However, the modification of the default self-attention computation and the use of $Q$, $K$, $V$ vectors that are extracted from three distinct images create a domain gap that needs to be addressed to enhance the quality of the prediction. Hence, we adapt the *attention map contrasting, swap-guidance*, and *AdaIN* mechanisms employed by Alaluf et al. (2024).

*Attention map contrasting* supplements the function of the temperature hyperparameter ($\tau$), where it enhances the focus on the relevant regions (*i.e.* semantically similar regions) between $C$ and $A$, while attenuating the irrelevant (*i.e.* color similarity between unrelated objects). Additionally, this operation adjusts the scale of the values to suit the pre-trained Stable Diffusion pipeline. This contrasting operation can be expressed as,

$$\alpha_D^{(t)} = \mu(\alpha_D^{(t)}) + \beta \cdot \left(\alpha_D^{(t)} - \mu(\alpha_D^{(t)})\right),\tag{8}$$

where $\mu$ is the mean operation and $\beta$ is a hyperparameter that controls the scale.

*Swap-guidance* is derived from classifier-free diffusion guidance implementation proposed by Ho & Salimans (2022). Given the noise prediction from the default self-attention formulation ($\eta_{default}^{(t)}$) and the noise prediction from our modified formulation ($\eta_{modified}^{(t)}$), the final noise prediction ($\eta^{(t)}$) can be expressed as,

$$\eta^{(t)} = \eta_{default}^{(t)} + \frac{\gamma \cdot (T-t)}{T} \cdot \left(\eta_{modified}^{(t)} - \eta_{default}^{(t)}\right),\tag{9}$$

where $\gamma$ is a hyperparameter controlling the scale. This mechanism gradually incorporates the modified noise prediction as the denoising progresses, which directs the denoising process through denser regions of the Stable Diffusion's generative pipeline, thus alleviating unwarranted artifacts.

*Adaptive instance normalization (AdaIN)*, which was proposed by Huang & Belongie (2017) is used to align the color distribution between the prediction ($D$), which is initialized using the noise space of the query image ($C$), and the final ground-truth/task color space (*i.e.* color space of $B$). AdaIN is implemented as follows,

$$z_D^{(t)} \leftarrow \text{AdaIN}(z_D^{(t)}, z_B^{(t)}), \tag{10}$$

$$z_D^{(t)} \leftarrow \frac{z_D^{(t)} - \mu(z_D^{(t)})}{\sigma(z_D^{(t)})} \cdot \sigma(z_B^{(t)}) + \mu(z_B^{(t)}), \tag{11}$$

where $\mu$ and $\sigma$ correspond to the mean and standard deviation operations.

### 2.3 IMPLICITLY-WEIGHTED PROMPT ENSEMBLING (IWPE)

Bar et al. (2022); Xu et al. (2023) demonstrate that providing the model with multiple source-target example prompts improves the prediction performance of visual in-context learning. These methods ensemble multiple prompts at the input level by stitching together prompt images and corresponding ground-truths to form a grid-like structure in the form of a composite image. This composite image is then input to the model, which is expected to leverage information from multiple prompts to make a better prediction compared to the single prompt scenario. However, for a fixed-resolution foundation model, as the number of source-target examples increases, the effective size of each image in the grid is reduced. This leads to performance deterioration, mainly because of the loss of details in the input (Xu et al., 2023).

To address this issue, instead of ensembling prompts in the image space through a composite input, SegGPT (Wang et al., 2023b) proposed a feature space ensembling method. In this method, multiple prompts are processed in parallel and aggregated by averaging the features at the end of each attention layer. This approach assumes that all the example prompts are equally informative and they are uniformly weighted during the averaging process. However, such uniform weighting of prompts is sub-optimal since it prevents the model from benefiting from more relevant prompts. To this end, we propose a simple but effective ensembling method by integrating the ensembling into the attention computation, allowing the prompt patches to be implicitly weighted based on their correspondences with each query patch. Specifically, we concatenate each of the $K$ and $V$ vectors from each of the prompts prior to computing $\Delta\phi_D^{(t)}$. Specifically, for $n$ number of prompts, we re-write Eq. (7) as,

$$\Delta\phi_D^{(t)} = \text{softmax}\left(\frac{Q_C^{(t)} \cdot \left(\bigoplus_{i=1}^{n} K_{A_i}^{(t)}\right)^T}{\tau \cdot \sqrt{d}}\right) \cdot \left(\bigoplus_{i=1}^{n} V_{B_i}^{(t)}\right), \tag{12}$$

where $\oplus$ is the concatenation operation and $i$ is the index for each prompt.

## 3 EXPERIMENTS AND EVALUATIONS

To demonstrate the ability of the proposed approach to generalize across different tasks, we evaluate it on six downstream tasks. For all these tasks, we use the unsupervised prompt retrieval (Zhang et al., 2023) to select the candidates for the prompt images. Specifically, this method chooses the nearest neighbours of the query image as prompt candidates. The nearest neighbour retrieval is based on the cosine similarity of CLIP's vision encoder (Radford et al., 2021) embeddings between the query image and the example prompt images.

For a fair comparison, we use Visual Prompting (Bar et al., 2022) and IMProv (Xu et al., 2023) as baselines. Although these approaches involve a training step, unlike Wang et al. (2023a;c), they do not rely on curated or annotated data that come from related out-of-domain tasks. Additionally, since IMProv supports supplementary text guidance, we include results with and without text guidance in our comparisons. Further, we also evaluate and compare the effectiveness of multiple prompts. Please refer to the supplementary for details on datasets, evaluation metrics, and sensitivity analysis for different hyperparameters. For the results in the following subsections, we choose one set of hyperparameters that provided optimal performance for all the tasks.

Table 1: Quantitative performance comparison of the proposed approach with recent approaches on foreground segmentation and single object detection using the Pascal-5i dataset.

| Model | Foreground Segmentation | | | | (mIoU ↑) | Single Object Detection (mIoU ↑) | | | | |
|---|---|---|---|---|---|---|---|---|---|---|
| | Split 0 | Split 1 | Split 2 | Split 3 | Avg. | Split 0 | Split 1 | Split 2 | Split 3 | Avg. |
| Number of Example Prompts: 1 | | | | | | | | | | |
| Visual Prompting | 34.85 | 38.55 | 34.51 | 32.24 | 35.04 | 48.82 | 48.52 | 45.11 | 42.72 | 46.29 |
| IMProv (w/o text) | 41.46 | 43.60 | 39.70 | 33.22 | 39.50 | 46.10 | 47.26 | 41.97 | 39.96 | 43.82 |
| IMProv (w/ text) | 41.31 | 44.64 | 40.86 | 35.93 | 40.69 | 44.69 | 48.10 | 44.53 | 40.34 | 44.42 |
| SD-VICL (ours) | **44.05** | **45.17** | **44.36** | **42.11** | **43.92** | **54.45** | **52.92** | **51.56** | **47.27** | **51.55** |
| Number of Example Prompts: 5 | | | | | | | | | | |
| Visual Prompting | 36.70 | 40.02 | 36.18 | 32.56 | 36.37 | 51.59 | 49.30 | 46.80 | 44.66 | 48.09 |
| SD-VICL (ours) | **55.55** | **56.08** | **55.84** | **54.49** | **55.49** | **58.99** | **56.31** | **57.09** | **56.01** | **57.10** |

Table 2: Quantitative performance comparison of the proposed approach with recent approaches on semantic segmentation, keypoint detection, edge detection, and colorization.

| Model | Semantic Segmentation (Cityscapes) | | Keypoint Detection (DeepFashion) | | Edge Detection (NYUDv2) | | Colorization (ImageNet) | |
|---|---|---|---|---|---|---|---|---|
| | mIoU ↑ | Acc. ↑ | MSE ↓ | PCK ↑ | MSE ↓ | LPIPS ↓ | LPIPS ↓ | FID ↓ |
| Number of Example Prompts: 1 | | | | | | | | |
| Visual Prompting | 21.30 | 71.52 | 35.83 | 10.87 | 0.1006 | 0.3925 | 0.4166 | 111.06 |
| IMProv (w/o text) | 15.97 | 68.44 | 31.30 | 18.38 | 0.1059 | 0.4278 | 0.3895 | 104.74 |
| IMProv (w/ text) | 15.80 | 67.88 | 44.67 | 2.81 | 0.1125 | 0.5146 | 0.3899 | 105.90 |
| SD-VICL (ours) | **22.08** | **74.93** | **5.36** | **77.19** | **0.0282** | **0.1548** | **0.2810** | **53.40** |
| Number of Example Prompts: 5 | | | | | | | | |
| Visual Prompting | 21.70 | 70.95 | 36.57 | 10.32 | 0.1000 | 0.3900 | 0.4156 | 112.91 |
| SD-VICL (ours) | **27.01** | **81.75** | **4.37** | **82.24** | **0.0213** | **0.1216** | **0.2272** | **44.84** |

## 3.1 RESULTS AND ANALYSIS

Quantitative metrics for all the six tasks are reported in Tabs. 1 and 2. The proposed approach consistently outperforms existing uncurated based approaches in all the tasks.

**Foreground segmentation**: With a single prompt, the proposed method achieves absolute improvements of 8.9% and 3.2% in *mean intersection over union* (mIoU) on an average across all splits in the Pascal-5i dataset (Shaban et al., 2017), compared to Visual Prompting and IMProv, respectively. The proposed ensembling method with five prompts using the re-purposed Stable Diffusion model further improves the mIoU by 11.6% (in absolute terms) as compared to the single prompt case.

**Single object detection**: Similar to foreground segmentation, with a single prompt, the proposed method improves the mIoU by 5.3% and 7.1% (in absolute terms) on average across all splits in the Pascal-5i dataset (Shaban et al., 2017) as compared to Visual Prompting and IMProv, respectively. The use of multiple prompts through implicitly-weighted prompt ensembling results in an additional 5.6% of improvement as compared to the single prompt case.

**Semantic segmentation**: With a single prompt on the Cityscapes dataset (Cordts et al., 2016), our method results in an absolute improvement of 0.8% in mIoU and 3.4% increase in accuracy as compared to Visual Prompting. Similarly, we observe an absolute gain of 6.1% in mIoU and a 6.5% in accuracy as compared to IMProv. With IWPE, the mIoU and accuracy further improve by 4.9% and 6.8% (in absolute terms) as compared to the single prompt based approach. The proposed approach with a single prompt in the case of semantic segmentation achieves lower performance improvements over existing methods as compared to other tasks like foreground segmentation and keypoint detection. However, with additional prompts, the proposed method is able to achieve much higher gains similar to what is observed in other tasks. This performance boost can be attributed to two key factors: (1) multiple prompts help resolve conflicts between similar or overlapping classes by mitigating inter-class attention ambiguities, and (2) a single prompt is unlikely to contain all classes that are potentially present in the query image of a multi-class task and in such cases, access to multiple prompts increases the probability that all relevant classes are available to the model for making accurate predictions.

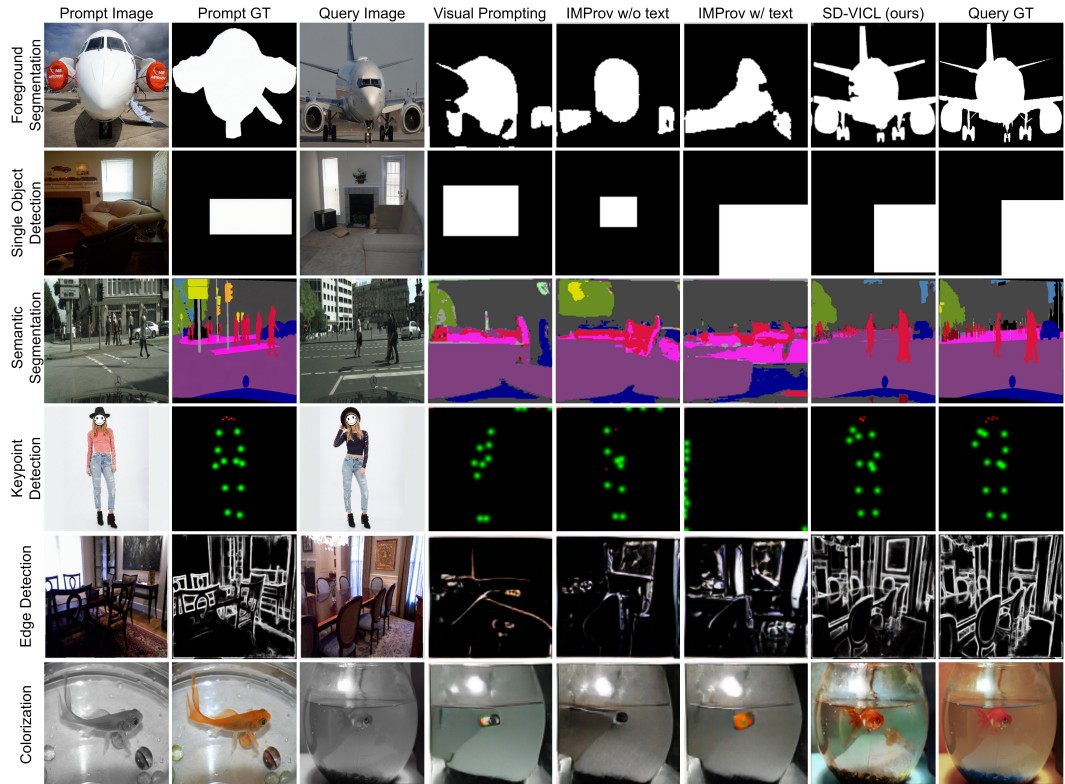

Figure 4: Qualitative evaluation, where we compare the performance of Visual Prompting (Bar et al., 2022) and IMProv (Xu et al., 2023) with our proposed method on six different tasks. It can be seen that our method produces visually superior results as compared to the baselines.

**Keypoint detection**: When evaluated on the DeepFashion dataset (Liu et al., 2016), our method demonstrates a substantial improvement, reducing the MSE by $6\times$ while achieving a $7\times$ gain in *percentage of correct keypoints* (PCK), compared to the best-performing baseline. Furthermore, the use of five prompts enhances these metrics, yielding additional relative gains of 18.5% for MSE and 6.5% for PCK.

**Edge detection**: On the NYUDv2 dataset (Silberman et al., 2012), our approach reduces the mean square error (MSE) by 72.0% and LPIPS (Zhang et al., 2018) by 60.6% as compared to the best performing existing approach. Note that these metrics improve an additional 24.5% and 21.4%, respectively, with our prompt ensembling method.

**Colorization**: Evaluating on the ImageNet dataset (Russakovsky et al., 2015), our model reduces LPIPS by 27.9% and FID (Heusel et al., 2017) by 49.0%, relative to IMProv, which yields the next best performance, and shows further gains of 19.1% and 16.0% respectively, with five prompts.

Overall, our method outperforms both Visual Prompting and IMProv across all tasks by considerable margins. Additionally, the integration of multiple prompts using IWPE consistently yields performance improvements over the single-prompt scenario.

Supplementing the quantitative results, we also demonstrate visual comparisons of the outputs for each task in Fig. 4[3], further highlighting the superior performance of our method over baselines. We observe that while both Visual Prompting and IMProv perform reasonably in single object detection and foreground segmentation, they show a weaker performance in tasks such as colorization, edge detection, and semantic segmentation, where capturing fine details and preserving structural information of the query is crucial. Moreover, both baselines notably struggle with keypoint detection, whereas the proposed approach demonstrates far superior results, indicating its robustness to different out-of-domain tasks.

---

[3]Additional qualitative results for each task can be found in the supplementary.

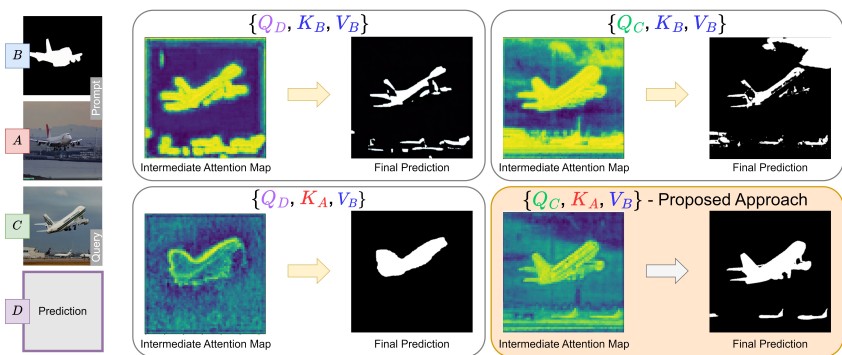

Figure 5: Qualitative examples of alternative attention formulations.

## 3.2 ABLATION STUDY

In this section, we present the details of the different ablations that we conducted to demonstrate the significance of different design choices like the attention formulation, effectiveness of implicitly-weighted prompt ensembling, and the impact of number of prompts on the performance. We refer the reader to the supplementary material for a detailed analysis on sensitivity of the proposed approach to different hyperparameters like attention temperature, resolution of the self-attention layers, contrastive strength parameter, swap-guidance scale, and AdaIN.

**Alternative attention formulations:** With regards to the attention formulation between query and the prompt, there are potentially multiple variants that could be used instead of the one described by Eq. (7). These candidate formulations can be derived by substituting the $Q$ and $K$ of Eq. (7) with the corresponding elements of each of these sets: $\{Q_D, K_B\}$, $\{Q_C, K_B\}$, and $\{Q_D, K_A\}$. Since the prediction needs to correspond to the features of the prompt ground-truth, the value vector, $V$, needs to come from $B$ and cannot be substituted with other alternate options. Fig. 5 illustrates a subset of these variants along with the predictions obtained using these alternate formulations. The quantitative performance corresponding to these candidate formulations are presented in Tab. 3. However, since the prompt ground-truth lacks semantics, such formulations (*i.e.* $\{Q_D, K_B, V_B\}$, $\{Q_C, K_B, V_B\}$) tend to focus on color similarities rather than inferring the underlying semantic correlations.

Alternatively, we can formulate the attention using the Query vector from the prediction ($D$) itself, similar to the approach followed by Alaluf et al. (2024). In this scenario, the intermediate predictions at early denoising stages closely resemble those produced by our formulation. However, in later denoising stages, the performance deteriorates as the prediction gradually shifts towards the prompt ground-truth, which lacks semantics, impairing the prediction performance. As seen in Fig. 5 and Tab. 3, the proposed formulation demonstrates superior performance which is achieved by ensuring that at each denoising step, the process is guided by the query and prompt latents at the corresponding denoising stages, thereby preserving the essential semantics needed for better context and task inference.

Table 3: Ablation of attention formulations on foreground segmentation evaluated on Pascal-5i.

| Method | mIoU ↑ |
|---|---|
| $\{Q_C, K_B, V_B\}$ | 12.54 |
| $\{Q_D, K_B, V_B\}$ | 12.95 |
| $\{Q_D, K_A, V_B\}$ | 23.68 |
| Proposed: $\{Q_C, K_A, V_B\}$ | **55.49** |

**Effectiveness of implicitly-weighted prompt ensembling (IWPE):** Here, we comapre our proposed ensembling method, IWPE, to SegGPT's feature ensembling (FE) approach. For all tasks except semantic segmentation, we observed that both methods yielded similar gains over the single prompt scenario. In the case of semantic segmentation, IWPE demonstrated significantly better performance. As tabulated in Tab. 4, while FE yields an absolute improvement of merely 0.9% in mIoU and 0.9% in accuracy, IWPE achieves substantial gains, improving mIoU by 4.9% and accuracy by 6.8% over the single prompt case. This notable improvement is primarily due to the capability of our ensembling method to dynamically weigh each prompt based on its correspondence with the query image. In contrast, as discussed previously, the feature ensembling approach assumes that all prompts are equally informative, which is suboptimal, especially in the case of tasks involving multiple classes like semantic segmentation.

For instance, one prompt may provide a stronger correlation with a specific class, while another prompt, would have a weaker association with that class. Our method by implicitly weighing the influence of different prompts based on their relative correspondence to the query, better captures these nuances, resulting in a significant boost in performance compared to SegGPT's uniformly weighted feature ensembling. Therefore, while both FE and IWPE perform at par in simpler tasks where the prompts are equally informative, our method is more beneficial for tasks that involve complex multi-class structures.

Table 4: Ablation of prompt-ensembling evaluated on semantic segmentation (Cityscapes).

| # Prompts | Ensembling type | mIoU ↑ | Acc. ↑ |
|---|---|---|---|
| 1 | N/A | 22.08 | 74.93 |
| 5 | FE | 22.97 | 78.84 |
| 5 | IWPE | **27.01** | **81.75** |

**Impact of number of prompts:** Here, we investigate the effect of the number of prompts on the performance and the inference speed. As illustrated in Fig. 6a, the mIoU scores improve with an increasing number of prompts. This result can be attributed to the model's ability to effectively leverage multiple prompts to better infer the task and establish the correspondence between the query and the prompts. The additional prompts help reduce ambiguities that may arise from using a single prompt. However, these gains come at the cost of reduced inference speed; specifically using five prompts cuts the inference speed to approximately half that of a single prompt. Besides the prompts, the number of denoising steps in the Stable Diffusion model is a key factor impacting the inference speed. This led us to the question — *With the availability of multiple prompts, does the model require as many denoising steps as in the single prompt case?* In other words, we explored if we can trade-off a few denoising steps without impacting the performance when additional context is provided by multiple prompts. To this end, we conducted experiments where we evaluated the SD-VICL approach with different denoising steps. As shown in Fig. 6b, the mIoU for the five-prompt scenario at 30 denoising steps surpasses the maximum performance of the single-prompt scenario. Moreover, the maximum performance for the single-prompt case, achieved with 70 denoising steps, requires 231 seconds of inference time. In contrast, the five-prompt case achieves higher performance in just 159 seconds with 30 denoising steps[4].

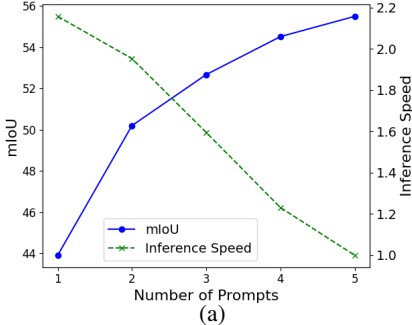
(a)

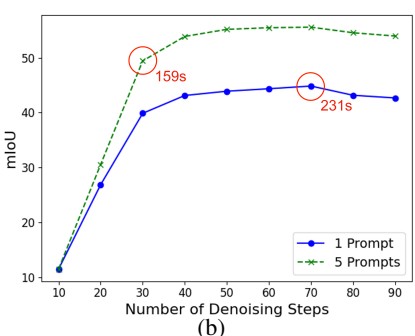
(b)

Figure 6: Effect of the number of prompts on the performance and inference speed[4]. While (a) depicts the variation in mIoU and the inference speed, with the number of prompts, (b) illustrates the variation in mIoU with the number of denoising steps for both single and five prompt cases.

## 4 CONCLUSION

In this paper, we propose a novel pipeline for visual in-context learning that explicitly incorporates the context between the query image and the prompt. Unlike existing visual in-context learning methods which rely on training or fine-tuning foundation models on curated or uncurated data to enable in-context learning, our approach is entirely training-free. Specifically, we introduce an in-place attention re-computation within the self-attention layers of an off-the-shelf Stable Diffusion model. Additionally, the proposed implicitly-weighted prompt ensembling technique facilitates effective integration of context through multiple prompts by implicitly weighing the prompts based on the relative correspondences. The versatility of the proposed method is demonstrated by its successful generalization across six different tasks, where it outperforms existing models that do not explicitly train on the tasks of interest nor use annotated samples related to the tasks.

---

[4]The inference times reported are based on evaluations conducted using a single V100 GPU.

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
