## A    SUPPLEMENTARY MATERIAL FOR "STABLE DIFFUSION MODELS ARE SECRETLY GOOD AT VISUAL IN-CONTEXT LEARNING"

This document is structured as follows:

- Appendix B: Related work

- Appendix C: Implementation details

- Appendix D: Additional quantitative results

- Appendix E: Discussion on training-based V-ICL models

- Appendix F: Additional ablations

- Appendix G: Limitations and future works

- Appendix H: Additional qualitative results

## B    RELATED WORK

In-context learning (ICL) has garnered significant attention in the field of natural language processing (NLP) with the advent of large-scale language models like GPT-3 (Brown et al., 2020) and its successors (Rae et al., 2021; Thoppilan et al., 2022; Chowdhery et al., 2023; Touvron et al., 2023). These models demonstrate the ability to perform tasks by conditioning on a small number of source-target examples, termed prompts, without any gradient updates or fine-tuning, effectively adapting to new tasks on-the-fly (Wei et al., 2022; Hao et al., 2022). The success of ICL in NLP has sparked interest in extending these capabilities to other domains, particularly in the realm of computer vision.

However, translating the concept of in-context learning from NLP to computer vision presents unique challenges due to the diversity in images and the inherent complexity of visual tasks. This has led to the emergence of two primary schools of thought in adapting ICL to computer vision (V-ICL).

The first approach adapts vision foundation models for in-context learning by training on uncurated datasets composed of random crops that potentially include examples of source images and corresponding targets (*e.g.* figures from computer vision papers). Research such as Visual Prompting (Bar et al., 2022) and IMProv (Xu et al., 2023) exemplify this approach, where they train a ViT-based MAE-VQGAN architecture (He et al., 2022; Esser et al., 2021) on the task of masked inpainting. During inference, these methods involve creating composite images by stitching together a query image with prompt examples, forming a grid-like structure with a placeholder mask for the prediction, that the inpainting model can process. While these methods yield promising results, this approach often suffers from weaker inference of context between the query image and prompt, lower resolution predictions, and overall weaker prediction quality.

The second school of thought aims to enhance prediction performance by training vision foundation models on curated/annotated datasets. This method involves training/finetuning a model but uses paired source-target images of multiple tasks as training data. Notable examples of this method include Painter (Wang et al., 2023a), Prompt Diffusion (Wang et al., 2023c), SegGPT (Wang et al., 2023b), Skeleton-In-Context (Wang et al., 2024), and Point-In-Context (Fang et al., 2024). While models such as Painter and Prompt Diffusion target a relatively diverse set of tasks, the others focus on building generalist models to cater specific tasks such as segmentation, skeleton sequence modeling, or 3D point cloud estimation. Although these models achieve improved results and provide important directions for future research for visual in-context learning, they require updating model weights using datasets related to the out-of-domain tasks. This in turn implies the need for training data on related out-of-domain tasks that we are trying to adapt to. We believe that this ideology diverges from the core principles of ICL as they often fall short in generalizing to novel tasks that are unrelated to the training set and rely on large annotated datasets. This approach, therefore, somewhat undermines the fundamental idea of ICL, which emphasizes the ability to adapt to new tasks without retraining nor requiring a large annotated dataset.

# C  IMPLEMENTATION DETAILS

**SD-VICL:** We base our experiments on an off-the-shelf Stable Diffusion model (Rombach et al., 2022), specifically the v1.5 checkpoint. Unless specified otherwise, we use the following hyper-parameters for all our evaluations: denoising time steps ($T$) = 70, attention temperature ($\tau$) = 0.4, contrast strength ($\beta$) = 1.67, and swap-guidance scale ($\gamma$) = 3.5. Further, we set the text condition of the Stable Diffusion pipeline to an empty string, and thus, no supplementary guidance is provided beyond the input prompts.

**Comparison baselines:** We use the publicly available repositories and checkpoints for both Visual Prompting (Bar et al., 2022) and IMProv (Xu et al., 2023) to generate the results for all the experiments. For the text-guided variant of IMProv, as specified in their paper, we provide the model with a string comprising of the location and task information (*e.g.* "Left - input image, right - Black and white foreground/background segmentation"). To ensure a fair comparison, all methods, including ours, are evaluated using the same set of prompts, which we obtain using the unsupervised prompt retrieval method outlined by Zhang et al. (2023).

**Tasks and datasets:** Below, we provide details on the tasks and datasets used for evaluations in our experiments:

- **Foreground segmentation:** This is a binary segmentation task, which predicts a binary mask of the object of interest (*i.e.* foreground) in an image. The prompt ground-truth is a black-and-white image with the foreground being white and the background being black. For evaluation, we use the Pascal-5i dataset (Shaban et al., 2017), which comprises of 1864 images belonging to 20 object classes. The images are divided into four splits, where each split consists of five unique classes. We use the mean intersection-over-union (mIoU) as the evaluation metric.

- **Single object detection:** This task is similar to the foreground segmentation task, however, in this task, the bounding box of the object of interest is predicted instead of the mask with the exact boundary. For this task, the prompt ground-truth is a black-and-white image with the bounding box colored in white. We use the same dataset as foreground segmentation but include only images with single instances of objects following Bar et al. (2022); Xu et al. (2023). The subset thus chosen consists of 1312 images and we report the mIoU scores.

- **Semantic segmentation:** This task predicts the per-pixel semantic label of a given image. We follow the method proposed by Wang et al. (2023a) to compose the prompt ground-truth, which assigns equally-spaced unique colors to each class. We use the Cityscapes dataset (Cordts et al., 2016), which consists of 19 classes (excluding the void classes), and the COCOStuff dataset (Lin et al., 2014), which consists of 27 mid-level classes. We report the mIoU and pixel accuracy scores as evaluation metrics.

- **Keypoint detection:** The task of keypoint detection entails locating the critical points or landmarks of an object. In this study, we focus on human pose keypoint detection, which predicts the locations of the 17 keypoints defined in COCO (Lin et al., 2014). Since the prompt ground-truth needs to be in the form of an image, we create an image that depicts the keypoints in the form of a heatmap as shown in Fig. 4. Each heatmap is created by superimposing Gaussian distributions centered on each keypoint. To accommodate the different spatial scales, we apply Gaussians with smaller variance for facial keypoints, which are relatively finer, and larger variance for body keypoints. These are visualized in two color channels: red for facial keypoints and green for body keypoints, facilitating easier decoding. For evaluation, following Hedlin et al. (2024), we use the DeepFashion dataset (Liu et al., 2016) and report metrics: MSE and the percentage of correct keypoints (PCK).

- **Edge detection:** The goal of this task is to predict the boundaries and edges within an image. For evaluation, we utilize the validation set of the NYUDv2 dataset (Silberman et al., 2012) comprising 654 images. Since the validation set did not have the ground-truth, we used the soft edge maps generated using HED (Xie & Tu, 2015) as the pseudo-ground-truth. For evaluation we compute the mean squared error (MSE) and the LPIPS loss (Zhang et al., 2018) between the HED-predicted edge map and the ICL predictions.

Table 5: Quantitative evaluation of single object detection on a subset of the Pascal-5i dataset, where larger objects with an area greater than 50% were excluded.

| Model | Single Object Detection (mIoU ↑) | | | |
|---|---|---|---|---|
| | Split 0 | Split 1 | Split 2 | Split 3 |
| Number of Example Prompts: 1 | | | | |
| Visual Prompting | 42.94 | 35.02 | 37.77 | 32.76 |
| IMProv (w/o text) | 42.32 | 36.52 | 36.32 | 31.83 |
| IMProv (w/ text) | 40.61 | 35.79 | 38.74 | 32.55 |
| Ours | **47.74** | **39.86** | **44.93** | **37.92** |
| Number of Example Prompts: 5 | | | | |
| Visual Prompting | 45.07 | 34.86 | 38.37 | 34.23 |
| Ours | **51.74** | **43.15** | **50.23** | **47.20** |

Table 6: Quantitative evaluation of semantic segmentation on the COCOStuff dataset.

| Model | Semantic Segmentation | |
|---|---|---|
| | mIoU ↑ | Acc. ↑ |
| Number of Example Prompts: 1 | | |
| Visual Prompting | 15.31 | 39.07 |
| IMProv (w/o text) | 17.09 | 41.64 |
| IMProv (w/ text) | 17.19 | 42.35 |
| Ours | **28.32** | **56.84** |
| Number of Example Prompts: 5 | | |
| Visual Prompting | 13.01 | 36.12 |
| Ours | **21.80** | **53.01** |

- **Colorization:** In this task, the objective is to colorize a given grayscale image. Similar to Bar et al. (2022); Xu et al. (2023) we randomly sample 1000 images from the validation set of ImageNet (Russakovsky et al., 2015) for evaluation. We compute the LPIPS loss and the FID score (Heusel et al., 2017) between the original colored image and the colorized prediction to evaluate the perceptual similarity.

## D  ADDITIONAL QUANTITATIVE RESULTS

In Tab. 1, we present the results of single object detection evaluated on the entire dataset for a more generalized assessment. However, in Tab. 5, we follow the approach of Bar et al. (2022) and evaluate single object detection on a subset of the Pascal-5i dataset, where images with objects covering more than 50% of the image are excluded. While we observe an overall drop in absolute scores for all methods, the performance trends remain consistent with Tab. 1. This decline in performance can be attributed to the fact that larger objects are generally easier to detect than smaller ones, as noted by Bar et al. (2022) as well.

Furthermore, we evaluated semantic segmentation on the COCOStuff dataset (Lin et al., 2014), where we report the results in Tab. 6. In contrast to the trend observed in Tab. 2, where performance improved with five example prompts compared to the single prompt, we could see a performance deterioration with five prompts in this case. Upon analysis, we identified that this performance decline was caused by the inconsistencies in labeling within the dataset, which creates confusion when inferring the context with multiple prompts, thereby negatively impacting the results.

## E  DISCUSSION ON TRAINING-BASED V-ICL MODELS

As highlighted in the main paper, we are the first to propose a training-free paradigm that *uncovers* the V-ICL properties of a vision foundation model. For fairness, in Sec. 3.1 of the main paper, we compared our results against Visual Prompting (Bar et al., 2022) and IMProv (Xu et al., 2023), as these models are the closest to our approach. Despite incorporating a training step, these models

Table 7: Extended quantitative evaluations against training-based V-ICL models, Painter (Wang et al., 2023a) and LVM (Bai et al., 2024)

| Model | FG Seg. mIoU ↑ | Obj. Det. mIoU ↑ | Edge Detection MSE ↓ | Edge Detection LPIPS ↓ | Colorization LPIPS ↓ | Colorization FID ↓ |
|---|---|---|---|---|---|---|
| Painter | 55.09 | 54.28 | 0.0926 | 0.7294 | 0.3474 | 64.16 |
| LVM | 50.98 | 52.67 | 0.0499 | 0.4259 | 0.3142 | 56.40 |
| SD-VICL (ours) | **55.49** | **57.10** | **0.0213** | **0.1216** | **0.2272** | **44.84** |

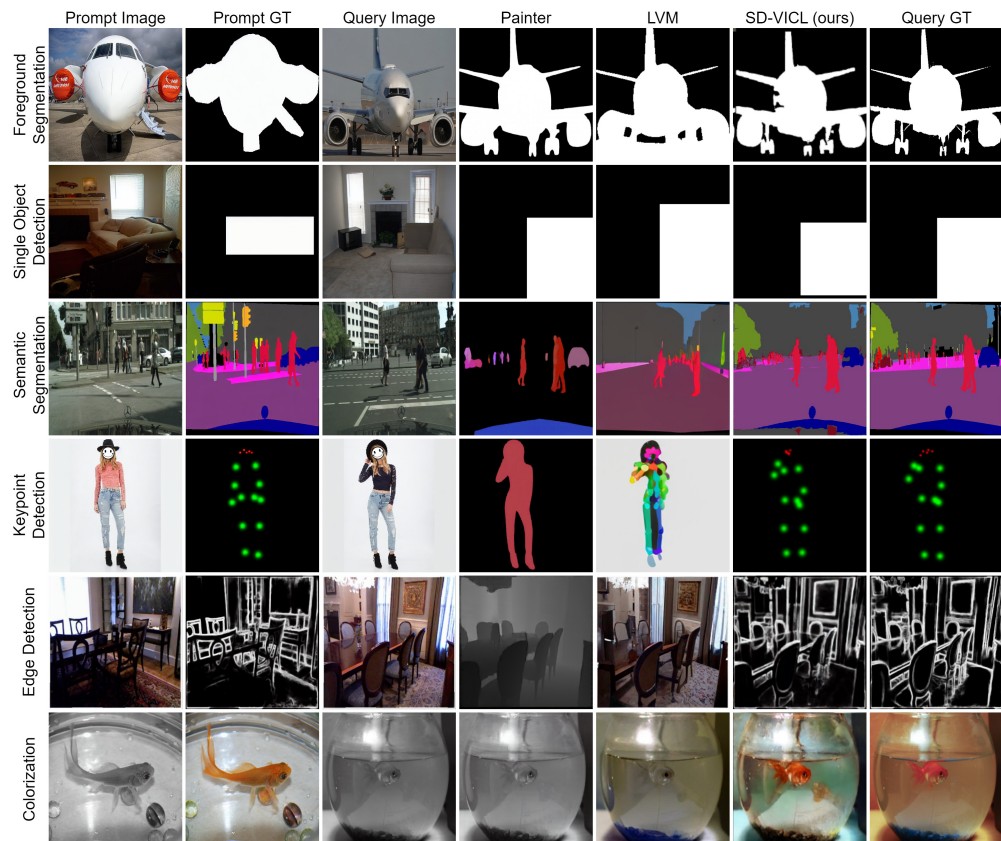

Figure 7: Additional qualitative comparisons illustrating the performance of training-based V-ICL models, Painter (Wang et al., 2023a) and LVM (Bai et al., 2024), against our proposed method on six different tasks. It can be seen that our method produces visually superior results as compared to the baselines.

are trained on uncurated datasets, unlike other models (Wang et al., 2023a;b) that rely on annotated data.

To ensure completeness, we extend our evaluations to training-based V-ICL models such as Painter (Wang et al., 2023a) and LVM (Bai et al., 2024). Painter utilizes a ViT-Large (Dosovitskiy et al., 2021) model trained on multiple annotated datasets (*e.g.* COCO (Lin et al., 2014), ADE20K (Zhou et al., 2019), and NYUv2 (Silberman et al., 2012)). On the other hand, LVM is based on OpenLLaMA's 7B model (Geng & Liu, 2023) and trained on the UVD-V1 (Bai et al., 2024) dataset, which comprises 50 datasets (*e.g.* LAION5B (Schuhmann et al., 2022)) containing annotated, unannotated, and sequence images.

The quantitative and qualitative comparisons are presented in Tab. 7 and Fig. 7, respectively. Overall, we could observe that our method outperforms both Painter and LVM across multiple tasks.

We observed that Painter and LVM often suffer from overfitting to training tasks, leading to poor generalization when exposed to novel tasks. Although visual in-context learning should ideally

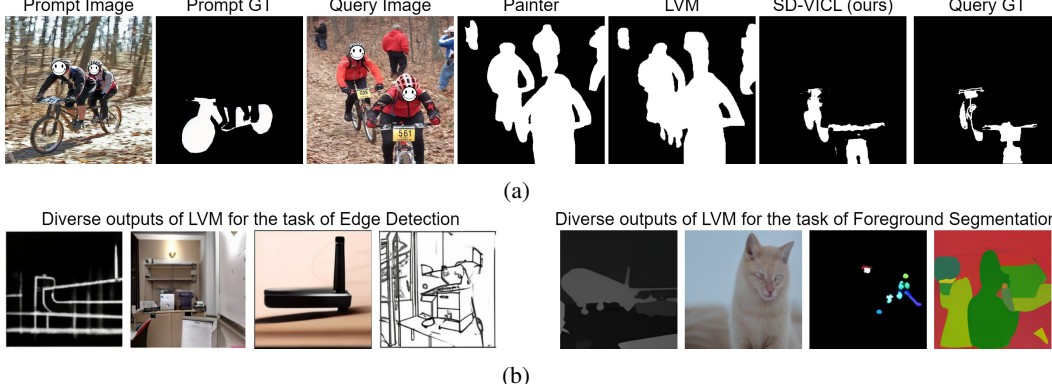

(a)

Diverse outputs of LVM for the task of Edge Detection          Diverse outputs of LVM for the task of Foreground Segmentation

(b)

Figure 8: **Failure cases of training-based V-ICL models**, Painter (Wang et al., 2023a) and LVM (Bai et al., 2024), implying poor task inference. In (a), both models fail in multi-class scenarios, segmenting the entire foreground instead of focusing on the region of interest defined by the prompt image and its corresponding ground truth. Examples in (b), depict inconsistent outputs generated by LVM for the same task (left: edge detection, right: foreground segmentation). The inputs for each of these outputs adhered to the same format as shown in Fig. 7, yet LVM produces outputs in diverse domains, deviating from the domain of the prompt groundtruth. These cases further emphasize the poor task inference capabilities of Painter and LVM.

infer the task from the relationship between the prompt image and its groundtruth, both models demonstrate weakness in this regard.

Painter performs well on simple tasks like foreground segmentation and object detection when the query image contains a single foreground category (Fig. 7, row 1). However, in multi-class scenarios (Fig. 8a), Painter segments the entire foreground rather than focusing on the specific region of interest defined by the relationship between the prompt image and its groundtruth. Further, overfitting to training tasks is evident in rows 3 and 4 of Fig. 7, where Painter outputs a segmentation map in semantic segmentation with a different color scheme than defined in the prompt groundtruth. Similarly, for keypoint detection, Painter outputs a segmentation map instead of a heatmap for keypoints. Moreover, Painter struggles with colorization, often outputting the grayscale image itself. In edge detection, Painter outputs a depth map instead of the expected edge map (Fig. 7, row 2). This suggests overfitting to the NYUv2 dataset, where the edge map query/prompt images overlap with those used for depth estimation during their training.

Similar limitations are observed for LVM, including poor performance on multi-class foreground segmentation (Fig. 8a), overfitting to training tasks (Fig. 7, row 4), and lack of generalization. Additionally, LVM exhibits inconsistencies in its outputs, as shown in Fig. 8b. Specifically, for a given task, despite the format/domain of the inputs remaining unchanged, we observe that the generated outputs belong to diverse domains. For example, in foreground segmentation, while some outputs align with foreground segmentation, others unexpectedly belong to unrelated domains such as keypoints, segmentation maps, or RGB images. This inconsistency highlights LVM's inability to produce coherent predictions despite the task and input format remaining unchanged.

These observations highlight the limitations of Painter and LVM in task inference and context interpretation from input prompts. Their reliance on task-specific training data results in overfitting, leading to poor generalization on novel tasks. In contrast, our proposed training-free model demonstrates robust generalization and effective task inference, underscoring the benefits of uncovering V-ICL properties without additional training and the superiority of the proposed method to explicitly infer the context and task from the inputs, as intended by V-ICL.

# F ADDITIONAL ABLATIONS

In addition to the ablations discussed in of the main paper, we also experimented with the effects of several other factors: temperature hyperparameter, resolution of the self-attention layers, contrastive strength parameter, swap-guidance scale, and AdaIN.

**Temperature hyperparameter, $\tau$:** As shown in Eq. (7), we introduce a temperature hyperparameter ($\tau$) to the attention computation in order to control the sharpness of correspondence between the patches of the query image and the prompt image. While we use a constant temperature hyperparameter (*i.e.* $\tau = 0.4$) across all tasks to preserve generalization, we investigated the effect of $\tau$ on the performance of a few proxy tasks. We observed that the optimal temperature parameter varies notably with the task, which we depict in Fig. 9.

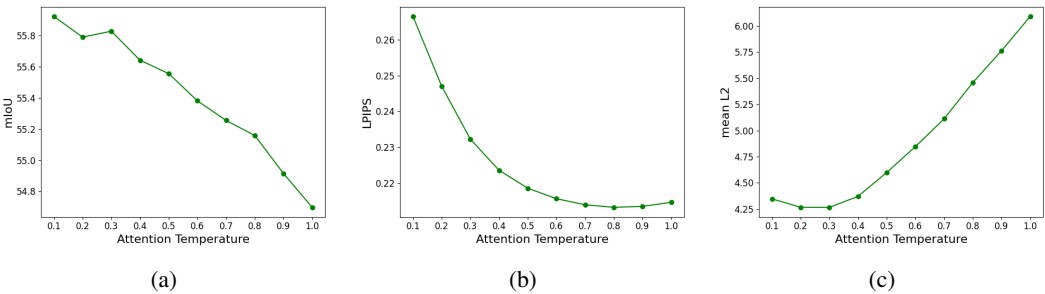

(a)       (b)       (c)

Figure 9: We illustrate the performance variation with respect to the attention temperature hyperparameter for the following tasks: (a) foreground segmentation, (b) colorization, and (c) keypoint detection.

**Contrast strength ($\beta$) and swap-guidance scale ($\gamma$) hyperparameters:** We adapt the *attention map contrasting* (Eq. (8)) and *swap-guidance* (Eq. (9)) methods from Alaluf et al. (2024) to address the domain gap introduced by using multiple images from different domains (*i.e.* source and target images belong to distinct domains). While we utilize the hyperparameter values proposed by Alaluf et al. (2024) (*i.e.* $\beta = 1.67, \gamma = 3.5$), we investigated their impact on performance using foreground segmentation as a proxy task. We depict the variation of the performance with respect to the contrast strength and the swap-guidance scale in Fig. 10. A notable improvement in performance could be observed with a contrast strength greater than 1.0 and with swap-guidance enabled.

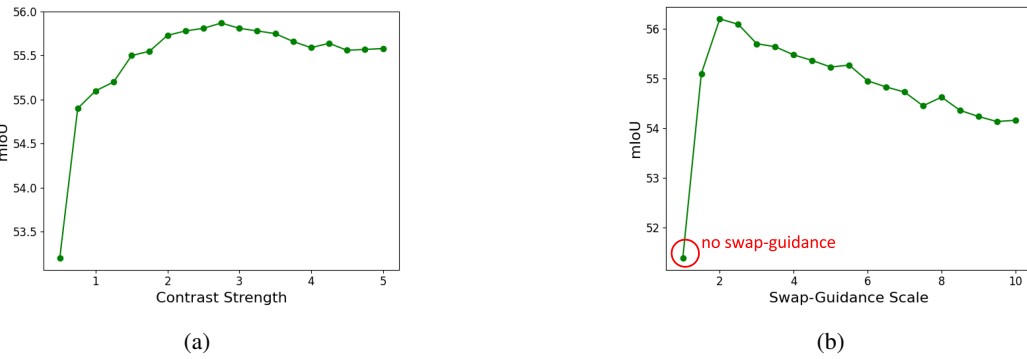

(a)              (b)

Figure 10: Performance variation with respect to (a) contrast strength and (b) swap-guidance scale hyperparameters.

**Adaptive instance normalization (AdaIN):** As explained in Sec. 2.2, we utilize AdaIN to align the color distribution between the prediction ($D$), which is initialized using the noise space of the query image ($C$), and the expected ground-truth color space (*i.e.* color space of $B$). In Fig. 11 we present a comparison example with and without AdaIN, and in Tab. 8 we tabulate the overall performance on foreground segmentation. A clear performance improvement could be observed with the incorporation of AdaIN.

**Resolution of attention layers:** The denoising U-Net in the Stable Diffusion pipeline contains self-attention layers at multiple resolutions: $16\times16$, $32\times32$, and $64\times64$. Consequently, we can apply the proposed in-place attention reformulation to any combination of these layers. We evaluated different combinations of these resolutions, with the results presented in Tab. 9. Additionally, we provide qualitative performance comparisons for each combination in Fig. 12. The best performance was

Table 8: Quantitative evaluation of with and without AdaIN evaluated using foreground segmentation.

| Model | mIoU ↑ |
|---|---|
| w/o AdaIN | 51.55 |
| w/ AdaIN | 55.49 |

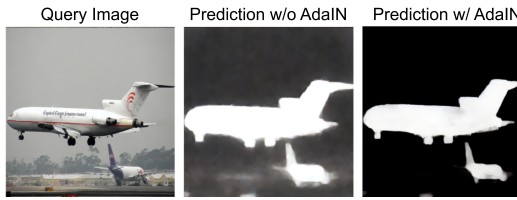

Figure 11: Example comparing the prediction with and without AdaIN.

Table 9: Quantitative evaluation on different combinations of resolutions which the self-attention layers could be modified using the proposed in-place attention reformulation.

| Resolution | | | mIoU ↑ |
|---|---|---|---|
| $16 \times 16$ | $32 \times 32$ | $64 \times 64$ | |
| ✓ | - | - | 11.39 |
| - | ✓ | - | 32.50 |
| - | - | ✓ | 50.33 |
| ✓ | ✓ | - | 35.48 |
| ✓ | - | ✓ | 52.52 |
| - | ✓ | ✓ | 53.76 |
| ✓ | ✓ | ✓ | **55.49** |

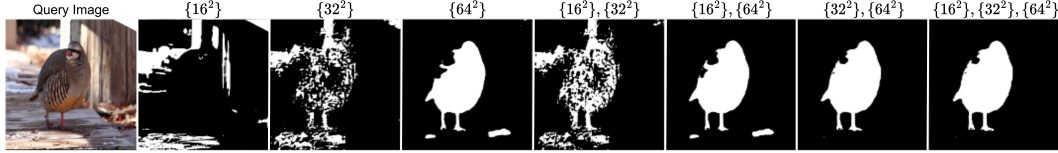

Figure 12: Qualitative examples of the output for each combination of self-attention layers modified using the proposed in-place attention reformulation.

achieved when modifying self-attention layers at all resolutions. This is intuitive, as it aggregates correspondences at multiple granularities, leading to a more comprehensive representation. In all our experiments, we use self-attention layers at all resolutions unless stated otherwise.

## G    LIMITATIONS AND FUTURE WORK

As with other diffusion-based methods, the primary limitation of our approach is the high inference time. Additionally, similar to other V-ICL methods, our model is sensitive to noisy prompts, as it relies only on a few prompts for context and task inference. Although the proposed implicitly-weighted prompt ensembling, along with the attention temperature, helps mitigate this sensitivity to a certain extent, there remains potential to further enhance robustness to noisy prompts. Moreover, scaling our method in the temporal dimension, by adapting to video generative models, could open up new possibilities, facilitating training-free V-ICL for video-based tasks. These directions could significantly expand the scope and applicability of visual in-context learning.

## H    ADDITIONAL QUALITATIVE RESULTS

We present additional qualitative examples for each task, foreground segmentation, single object detection, semantic segmentation, keypoint detection, edge detection, and colorization in Figs. 13 to 18 respectively.

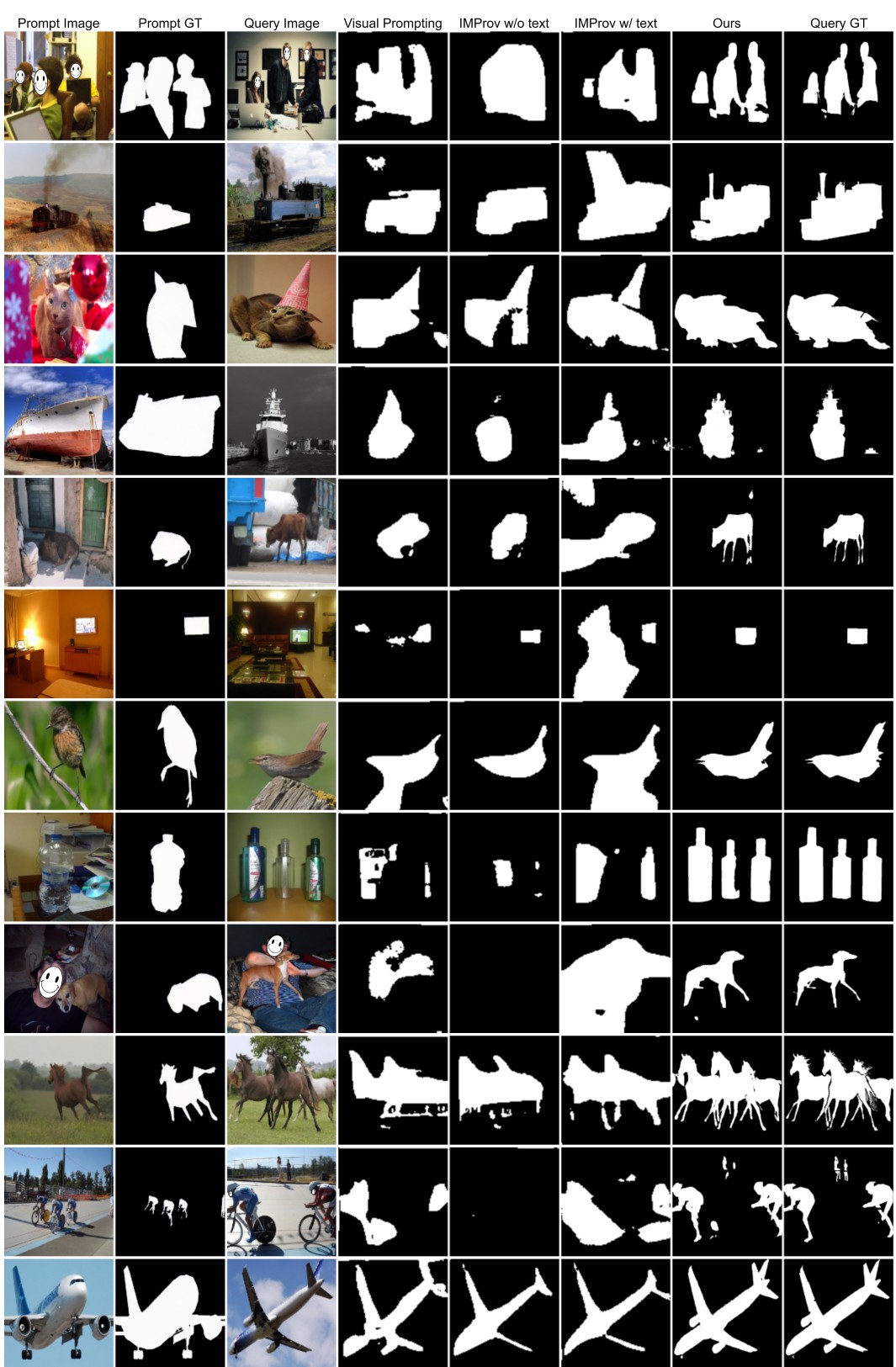

Figure 13: Qualitative examples of foreground segmentation in comparison with Visual Prompting (Bar et al., 2022) and IMProv (Xu et al., 2023).

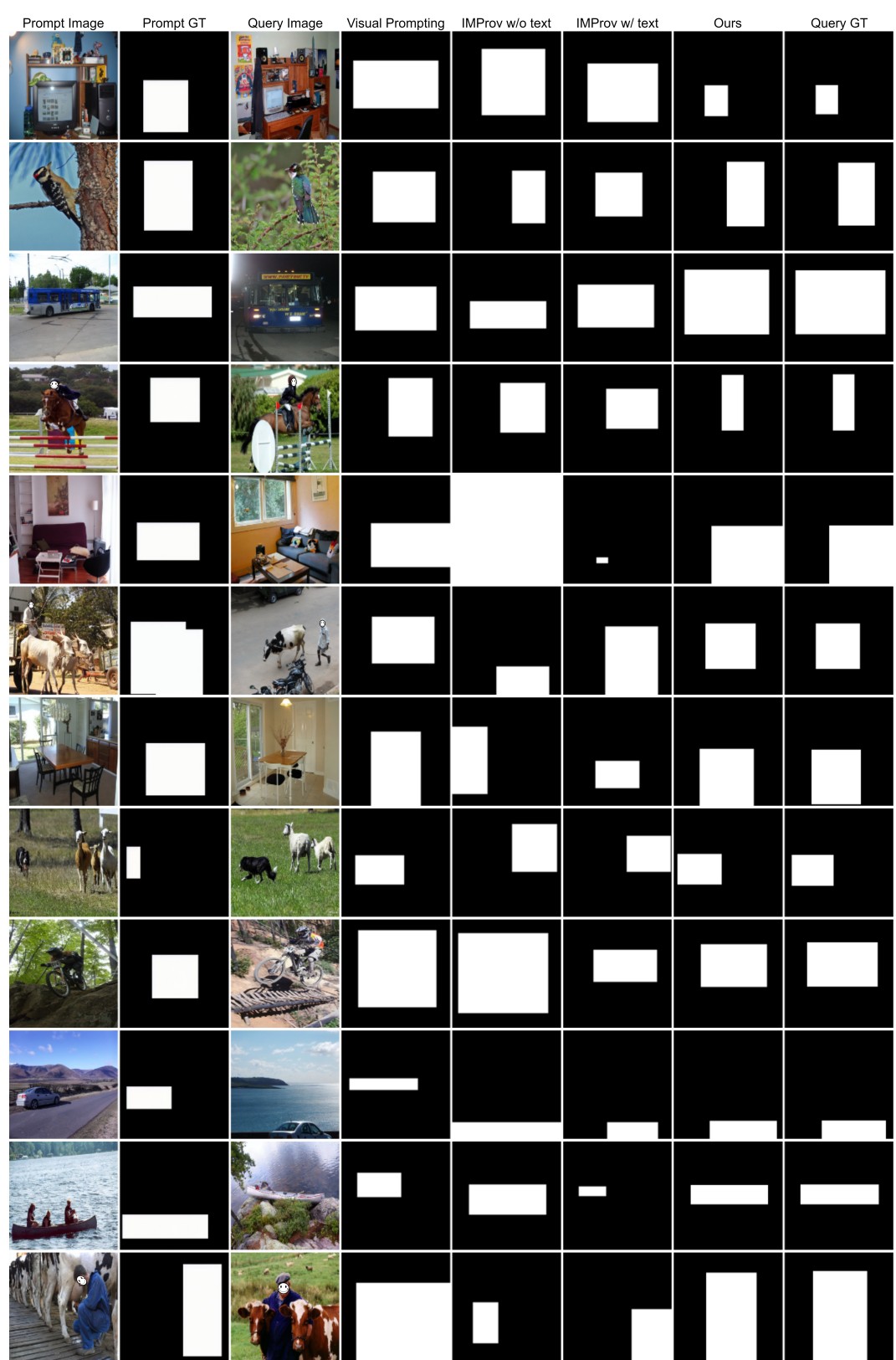

Figure 14: Qualitative examples of single object detection in comparison with Visual Prompting (Bar et al., 2022) and IMProv (Xu et al., 2023).

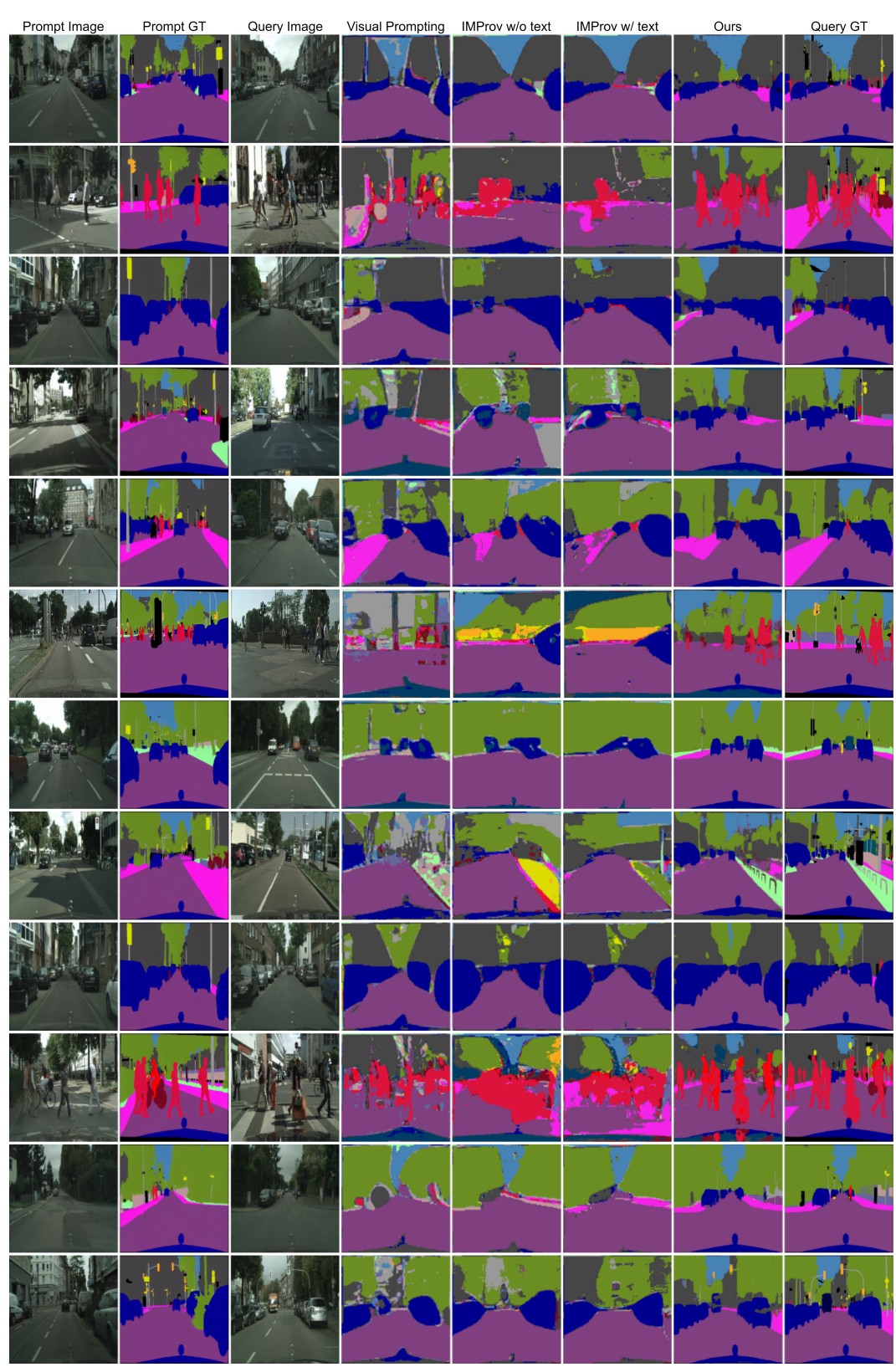

Figure 15: Qualitative examples of semantic segmentation in comparison with Visual Prompting (Bar et al., 2022) and IMProv (Xu et al., 2023).

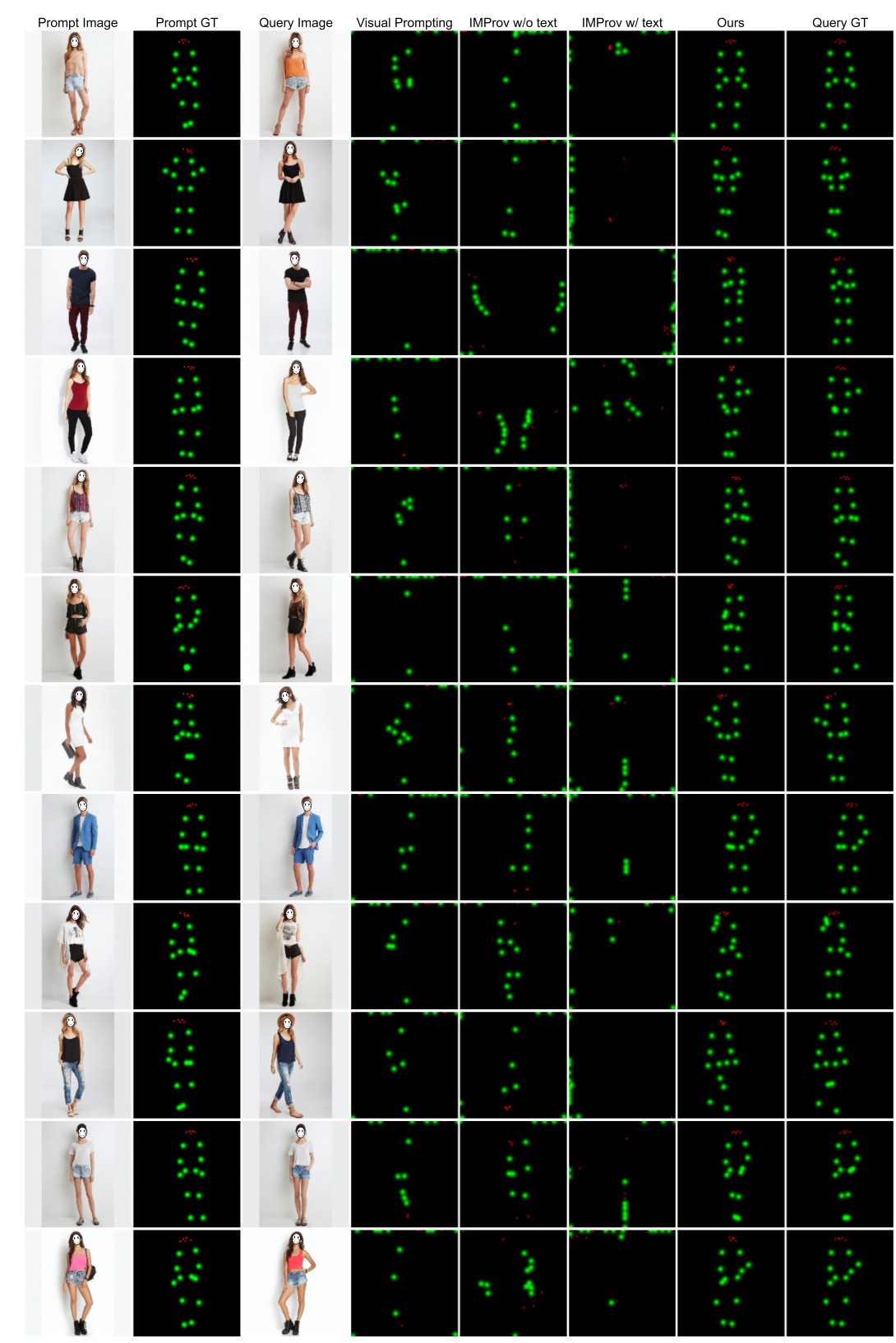

Figure 16: Qualitative examples of keypoint detection in comparison with Visual Prompting (Bar et al., 2022) and IMProv (Xu et al., 2023).

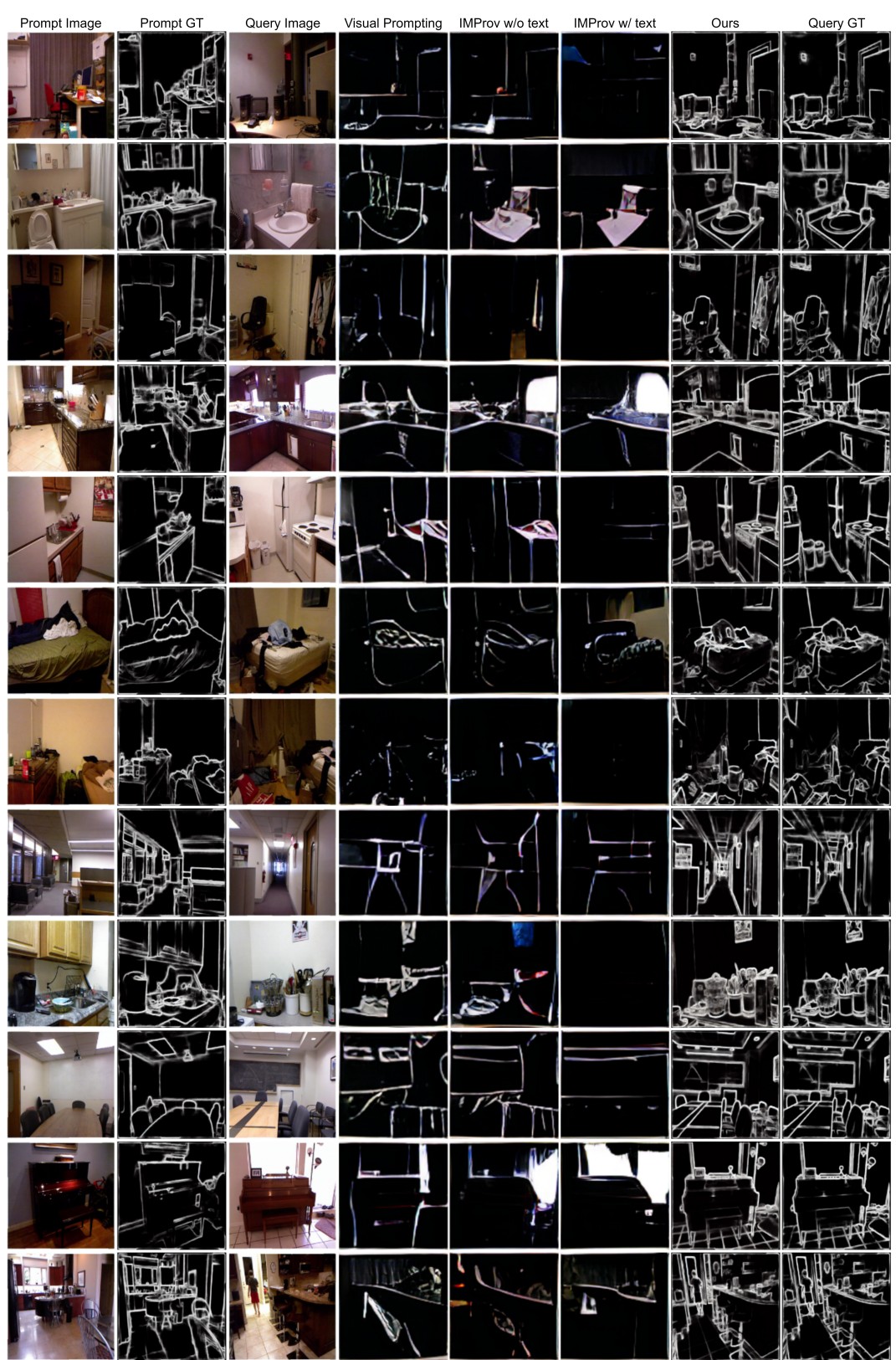

Figure 17: Qualitative examples of edge detection in comparison with Visual Prompting (Bar et al., 2022) and IMProv (Xu et al., 2023).

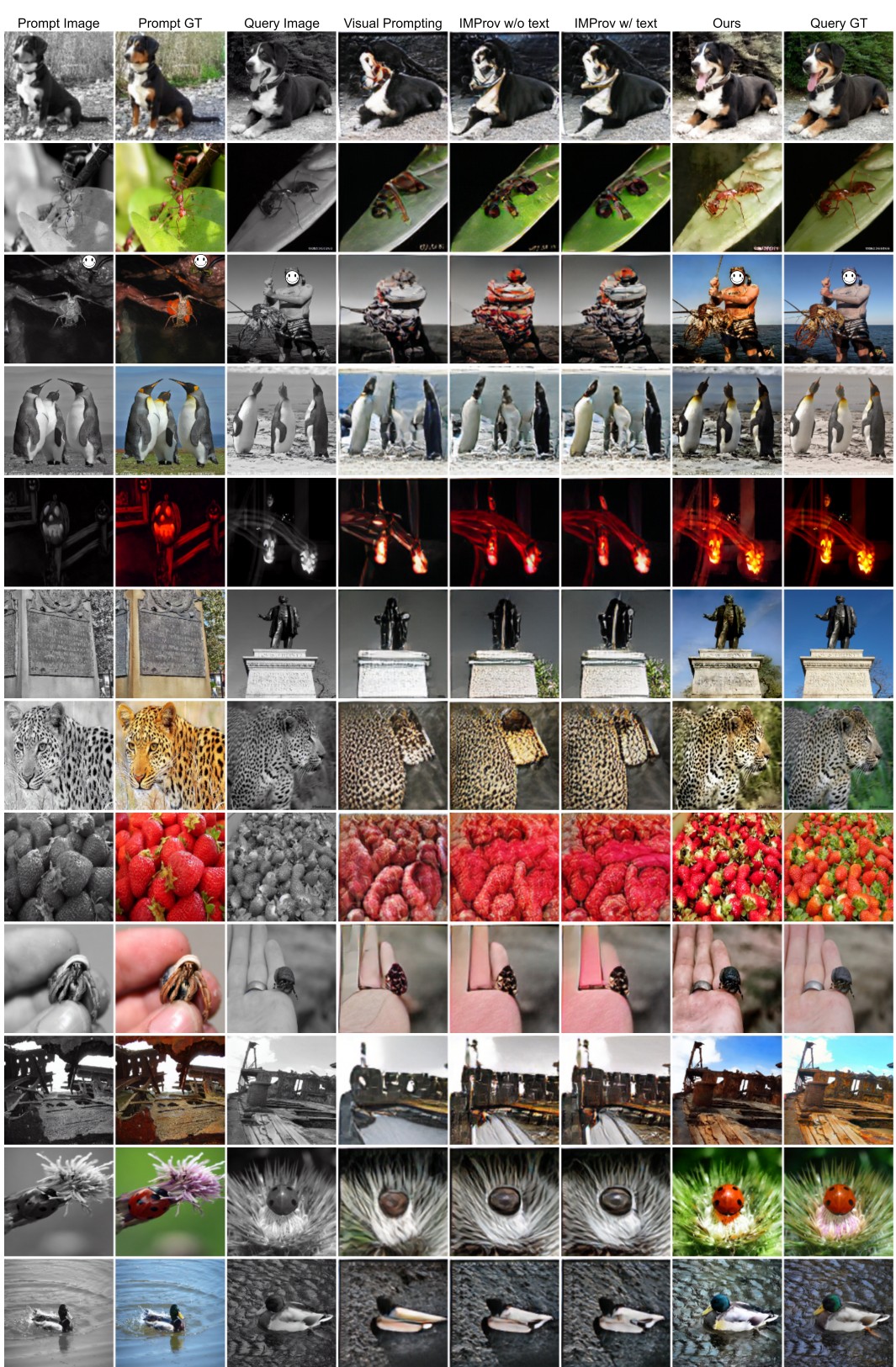

Figure 18: Qualitative examples of colorization in comparison with Visual Prompting (Bar et al., 2022) and IMProv (Xu et al., 2023).