# OpenReview forum: "STABLE DIFFUSION MODELS ARE SECRETLY GOOD AT VISUAL IN-CONTEXT LEARNING"
_ICLR.cc/2025/Conference — Submitted to ICLR 2025_

### Official Review · Reviewer_F13E · 2024-10-29

**Soundness:** 2
**Presentation:** 2
**Contribution:** 1
**Rating:** 5
**Confidence:** 5

**Summary:**

This paper shows that off-the-shelf Stable Diffusion models can be adapted for visual in-context learning (V-ICL) using in-place attention re-computation within self-attention layers. This method enables the model to leverage example prompts without fine-tuning and effectively adapts to six tasks, including segmentation and detection. Additionally, the approach utilizes multiple prompts through ensembling to improve task inference and overall performance. Experimental results show the effectiveness of this work.

**Strengths:**

1. The proposed method to perform the in-context inference is quite reasonable.
2. The studied topic is quite interesting, and I believe visual in-context learning is a critical issue.
3. The overall demonstration is good.

**Weaknesses:**

My major concern is the experimental parts, where I believe there many experiments shall be added.

1. The compared baseline methods are not enough. The author only made comparison with IMProv and MQ-VAE, How about SegGPT, Painter, and LVM? Besides, the results without specifically retrieval process are required to demonstrate the overall performance of the proposed methods. After all, this paper does not target on the design of best-demonstration selection. Finally, the selection-based methods (implemented based on MQ-VAE), such as UnsupPR, SupPR [1], prompt-SelF [2] and InMeMo [3] shall also be compared.

2. The second one goes to the generalization of this model. As shown in this paper, merely some discriminative tasks are made, where the major of them fall into the discriminative recognition task. I believe for visual ICL or ICL, the most importance point is the generalization. Therefore, more tasks, such as low-light enhancement and in-painting,  are encouraged (Please refer to Painter as a demonstration). Or alternatively, more advanced single-task could be implemented. For instance, SegGPT is able to Segment Anything with the given demonstration, in this way, this model targets on maximizing the potential of the segmenting task. I think either these trials could further demonstrate the effectiveness of the proposed method. In other words, in context for task-specific generalization (SegGPT), or for wide range-of representative tasks (Painter, LVM).

3. Fairly speaking, stable diffusion is not a pure Visual model instead of a multi-modal model. Therefore, I think more advanced multi-modal models could also be discussed, such as EMU. With the support of text, the in-context learning could be performed with more fancy operation. Regarding this aspect, the proposed method, emphasizing on visual side, is not implemented as expected since the usage of text seems to be overlooked.

[1] What Makes Good Examples for Visual In-Context Learning?

[2] Exploring effective factors for improving visual in-context learning.

[3] Instruct Me More! Random Prompting for Visual In-Context Learning

[4] Emu: Generative Pretraining in Multimodality

**Questions:**

None

---

> ### Author Response · Authors · 2024-11-25
> **Response to Reviewer F13E [1/3]**
>
> Thank you for the valuable feedback. In what follows, we address the reviewer's concerns in detail.
>
> &nbsp;
>
> ---
>
> ### **Limited comparisons**
>
> Since this concern is similar to the one raised by reviewer HFYR, we refer the reader to our comments in response to reviewer HFYR's comments (see Table 4 for quantitative comparisons). To summarize, we have added new experiments which show that the proposed approach outperforms additional recent methods such as Painter [1] and LVM [2]. Additionally, we also observed that these recent models show a tendency to overfit to the tasks/datasets that they are trained on and hence are limited in terms of generalization performance. This is in contrast to our approach, which is training free and has better generalization capabilities. (See Section E of revised supplementary for details and visualizations.) We further compare our method against task-specific unsupervised models that build on the Stable Diffusion priors as described in Tables 1 and 2 in the response to reviewer bhAU’s comments.
>
> &nbsp;
>
> ---
>
> ### **Paper does not target best-demonstration selection**
>
> We agree with the reviewer's observation about the scope of this work.  V-ICL research generally takes two broader directions:
> 1. Enabling in-context learning in vision foundation models [3,4].
> 2. Prompt engineering — finding the best prompt demonstrating the task [5-7].
>
> The scope of this work explores along the first direction.
>
> For prompt selection, we employed UnSupPR [5], which retrieves nearest neighbors based on CLIP features without task-specific training.  This preserves our training-free generalization, ensuring alignment with the V-ICL paradigm. In Table 5, we observe that the proposed method with unsupervised prompt retrieval performs better than other V-ICL approaches such as VPR-SupPR [5] and InMeMO [7] that use supervised prompt retrieval. Considering that these approaches show that the performance is enhanced by improving the prompt selection, we believe it applies to our approach as well, where better prompts can only improve the results further.
>
> #### Table 5: Comparison against prompt retrieval (PR) methods based on MAE-VQGAN, evaluated using foreground segmentation on Pascal-5i
>
> | Method            | Supervised PR? | mIoU ↑      |
> |--------------------|----------------|-------------|
> | VPR-SupPR         | ✔             | 35.56       |
> | InMeMo            | ✔             | 43.14       |
> | prompt-SelF       | ✘             | 41.02       |
> | Ours (1-prompt)   | ✘             | **43.92**   |
> | Ours (5-prompts)  | ✘             | **55.49**   |

---

> ### Author Response · Authors · 2024-11-25
> **Response to Reviewer F13E [2/3]**
>
> ---
>
> ### **Generalization of the model**
>
> We appreciate your suggestion to explore additional tasks to further demonstrate the model's generalization.
>
> Before addressing this, it is important to distinguish between:
> - **Generalist/multi-task models:** These models (e.g., [8-10]) are trained on multiple tasks and encode each task using unique embeddings or tokens. This often limits their generalization to unseen tasks.
> - **V-ICL models:** These models infer task and context *during inference* through the relationships between the query image, prompt image, and prompt groundtruth.
>
> We think that the Painter model is inclined towards multi-task models than V-ICL due to a few factors: (1) it is trained using annotated datasets (2) encodes the tasks using a color scheme in the prompt groundtruth, instead of inferring it from the relationship between the prompt image and its groundtruth during inference, (3) constrained to the tasks seen during training. In all fairness, the authors acknowledge that they generalize only to unseen categories of *seen tasks*, indicating that it has limited abilities to generalize to unseen tasks.  The fact that the tasks are encoded in the coloring scheme, and that Painter collapses to a multi-task model has been highlighted by Wang et al. in [11] - Sec 3.1.  This was observed in our experiments as well, where we saw that Painter struggles with novel tasks or different coloring schemes. For instance, different coloring in keypoint detection results in segmentation outputs, and edge detection using NYUv2 images results in depth maps, which implies poor task inference, overfitting to the training tasks/datasets, and dependence on the predefined coloring scheme (see Section E of revised supplementary for details and visualizations).
>
> Hence, while we agree that generalization to more tasks enhances the practicality of V-ICL, we emphasize that the proposed approach has better generalization ability due to the following reasons:
>
> - The proposed approach is able to perform better than other methods like Visual Prompting, IMProv, Painter, and LVM on six novel tasks.
>
> - We do not use a color scheme to encode the task, but rather model the relationship between prompt image and its groundtruth to infer the task, as intended in visual in-context learning.
>
> - Based on the reviewer's suggestion, we included an additional generative task of low-light enhancement (see Table 6) on the LOL dataset [12] where we show that the proposed method yields comparable results to the Painter network, despite the fact that Painter network uses the LOL dataset in training. With this expanded evaluation, we show the superiority of SD-VICL for a total of seven tasks, out of which two (colorization and low-light enhancement) are generative tasks.
>
> #### Table 6: Evaluation on low-light enhancement on LOL
>
> | Method          | PSNR ↑        | SSIM ↑       |
> |------------------|---------------|--------------|
> | Painter          | 22.40         | 0.872        |
> | SD-VICL (Ours)   | 23.25     | 0.848        |
>
>
> &nbsp;
>
> ---
>
>
> ### **Usage of text seems to be overlooked**
>
> While we agree that text descriptions could supplement in-context learning, in this work we exclusively focus on **visual** in-context learning, where we demonstrate that V-ICL properties could be uncovered from an off-the-shelf Stable Diffusion model, contrary to existing work which needs to train foundation models specifically targeting V-ICL. Hence, we intentionally did not use text conditioning to remove any language dependence for task inference, as ideally, V-ICL should infer the task and context from the prompt images. Additionally, we show in Tables 1 and 2 of the main paper that the proposed method without any text guidance is able to perform better than models like IMProv [4] that use text prompting in addition to visual prompting.
>
> Nonetheless, we acknowledge the importance of extending this paradigm to multi-modal contexts in future research. Based on the insights of our work, we hope future research will extend the proposed training-free paradigm to multi-modal prompting (e.g., images + text) and other multi-modal foundation models such as EMU [13], as highlighted by the reviewer.

---

> ### Author Response · Authors · 2024-11-25
> **Response to Reviewer F13E [3/3]**
>
> ---
>
> [1] Wang et al., "Images speak in images: A generalist painter for in-context visual learning" [CVPR'23]
>
> [2] Bai et al., "LVM: Sequential modeling enables scalable learning for large vision models" [CVPR'24]
>
> [3] Bar et al., "Visual prompting via image inpainting" [NeurIPS'22]
>
> [4] Xu et al., "IMProv: Inpainting-based Multimodal Prompting for Computer Vision Tasks" [arXiv'23]
>
> [5] Zhang et al., "What Makes Good Examples for Visual In-Context Learning?" [NeurIPS'23]
>
> [6] Sun et al., "Exploring effective factors for improving visual in-context learning" [arXiv'23]
>
> [7] Zhang et al., "Instruct me more! random prompting for visual in-context learning" [WACV'24]
>
> [8] Lu et al., "Unified-IO: A unified model for vision, language, and multi-modal tasks" [ICLR'22]
>
> [9] Wang et al., "OFA: Unifying architectures, tasks, and modalities through a simple sequence-to-sequence learning framework" [ICML'22]
>
> [10] Kolesnikov et al., "UViM: A unified modeling approach for vision with learned guiding codes" [NeurIPS'22]
>
> [11] Wang et al., "SegGPT: Segmenting everything in context" [ICCV'23]
>
> [12] Wei et al., "Deep Retinex Decomposition for Low-Light Enhancement" [BMVC'18]
>
> [13] Sun et al., "EMU: Generative pretraining in multimodality" [ICLR'24]

---

> > ### Comment · Reviewer_F13E · 2024-11-26
> >
> > Thanks for the authors' responses, and I think some of my concerns have been well addressed. However, I still have some concerns regarding the experimental parts. As the Reviewer bhAU mentioned, the prompt number is unknown in the update table. Besides, I notice you have designed a new ensemble method for multiple (5) prompt demonstration, how about its effectiveness on SegGPT and Painter? To ensure a fair comparison, I believe it is important to verify this ensemble mechanism effectiveness on other models. In this way, the comparison with Painter and LVM + this designed operation should also be implemented  Lastly, I think the claim regarding the non-use of text is quite strange. Since diffusion model is well-known for its text-to-image generative ability, but this work merely discuss the possibility of in-place key-value operation for pushing the newly input towards the target domain. This will diminish the use of SD model for ICL. For example, if we ignore the design operation within the denoising the unet, simply design a cross-attention-based feature operation, like constructing task vector, for VAE could also do the similar in-context learning as you done in this paper? With the support of relevant papers provided in Reviewer bhAU, such cross attention operation to implement in-context learning is quite a common. Therefore, I think the contribution of this paper shall be limited due to the limited exploration of SD on ICL.

---

> ### Author Response · Authors · 2024-12-02
> **Response to Reviewer F13E [1/2]**
>
> We appreciate the reviewer's thoughtful follow-up comments and the opportunity to clarify further. Below, we address each of the raised concerns in detail.
>
> &nbsp;
>
> ---
>
> ### **Clarification on the number of prompts in the updated tables**
>
> We apologize for not explicitly clarifying the number of prompts used in the updated table. As mentioned in our response to Reviewer bhAU (see https://openreview.net/forum?id=fKrFTGnoXY&noteId=x4d5tHpbN9), all results in the updated table were reported using five prompts, as this configuration effectively demonstrates the combined potential of our training-free V-ICL framework and the IWPE ensembling mechanism.
> Note that we have revised tables that now include results with a single prompt which still outperforms the related methods on both tasks.
>
> #### Table 1: Evaluation of semantic segmentation on Cityscapes
>
> | Method             | Eval. Method | mIoU ↑    | Acc. ↑    |
> |--------------------|--------------|-----------|-----------|
> | Tian et al., 2024 [1] | Hungarian    |  $21.2$      | $76.0$      |
> | Ours - 1 prompt | Hungarian    | $\underline{28.28}$ | $\underline{76.34}$ |
> | Ours - 5 prompts | Hungarian    | $\textbf{31.52}$ | $\textbf{82.72}$ |
>
> #### Table 2: Evaluation of keypoint detection on DeepFashion
>
> | Method             | MSE ↓       | PCK ↑     |
> |--------------------|-------------|-----------|
> | Hedlin et al., 2024 [2] | $6.46$      | $70.0$      |
> | Ours - 1 prompt | $\underline{5.36}$   | $\underline{77.19}$ |
> | Ours - 5 prompts | $\textbf{4.37}$   | $\textbf{82.24}$ |
>
> &nbsp;
>
> ---
>
> ### **Effectiveness of the proposed IWPE ensembling mechanism**
>
> Existing V-ICL models such as Painter and LVM are trained on curated datasets, relying on the model to implicitly learn the context and task relationships from the pre-defined input patterns provided. These methods, by design, use fixed input formats (e.g., composite images or sequences) that do not allow modifications to latent-based prompt ensembling or weighting methods, which we argue is suboptimal (Section 2.3 of the main paper). To clarify, adding IWPE to existing approaches would require re-training their models which is in contrast to the goal of our work to make visual in-context learning methods training-free.
>
> In contrast, our framework allows for flexibility in prompt weighting strategies. Hence, in Section 3.2 of the main paper, we evaluate our Implicitly Weighted Prompt Ensembling (IWPE) mechanism against the feature ensembling (FE) approach proposed by SegGPT, the only other latent-based ensembling method. IWPE shows substantial improvements over FE, enhancing mIoU by 4.0\% and accuracy by 2.9\%. These results highlight the effectiveness of IWPE in dynamically weighing prompts to improve task performance.
>
> &nbsp;
>
> ---
>
> ### **Clarification on not using text conditioning**
>
> We would like to clarify that the decision to focus exclusively on visual inputs in our work is deliberate and aligns with the goals of this paper. While Stable Diffusion is widely recognized for its text-to-image generative capabilities, our goal is to uncover its potential for **visual in-context learning (V-ICL)** using purely visual prompts, where the model infers task and context relationships solely from input-output pairs.
>
> This focus allows us to explore the feasibility of V-ICL without relying on text conditioning. Previous works, such as [14-16], have similarly omitted text conditioning of Stable Diffusion in their pipelines to demonstrate task-specific performance without dependence on textual descriptions. Additionally, the complexity of vision tasks often makes it challenging to design unambiguous text descriptions, further supporting our decision to rely solely on visual prompts.
>
> &nbsp;
>
> ---
>
> ### **Potential of cross-attention-based task vector in VAE-based models**
>
> We acknowledge the reviewer’s suggestion that simpler generative models like VAEs could potentially achieve similar ICL properties using cross-attention-based task vectors. However, our work demonstrates that Stable Diffusion’s iterative denoising mechanism is particularly advantageous for uncovering task and context relationships.
>
> Unlike in VAEs, the iterative nature of the denoising process gradually aligns the task and context relationships with the target domain, avoiding destabilization of the pre-trained generation process.
> Modifying an off-the-shelf foundation model to generate outputs in a new target domain requires mechanisms to smoothly integrate these changes into the generative pipeline, which the denoising process facilitates.
>
> While our insights could inspire future work to adapt similar techniques to VAE-based models, the iterative denoising framework provides a distinct advantage for the proposed training-free V-ICL pipeline.

---

> ### Author Response · Authors · 2024-12-02
> **Response to Reviewer F13E [2/2]**
>
> ---
>
> ### **Clarification on "*such cross attention operation to implement in-context learning is quite a common*''**
>
> We would like to clarify that the comment from Reviewer bhAU on the use of cross-attention being common was with regards to works like metric learning [17], few-shot segmentation [18,19], and multi-modal representation learning [20,21]. As it can be noted from Reviewer bhAU's latest comments, they acknowledge that the specific form of cross-attention that we use has not been explored earlier for visual in-context learning. As detailed in our response to Reviewer bhAU (see https://openreview.net/forum?id=fKrFTGnoXY&noteId=FoxGQ4LQZf), while the concept of cross-attention has been used in diverse tasks, our **specific formulation** differs significantly:
> - To the best of our knowledge, this specific form of cross-attention which models task and context relationships, is being utilized for the first time, particularly to facilitate in-context learning
>
> - Our method uses three distinct sources for the query, key, and value vectors (from the query image, prompt image, and prompt groundtruth, respectively). Unlike works such as TriBERT [21], which pairs key and value vectors within the same source, our formulation explicitly pairs keys and values from distinct images ($A$ and $B$) (further clarified in https://openreview.net/forum?id=fKrFTGnoXY&noteId=oGE8jLTFcA)
>
> - Existing works cited by Reviewer bhAU require further training after introducing cross-attention to capture correspondences between inputs. In contrast, our method operates in a completely training-free manner, as acknowledged by Reviewer bhAU.
>
> These distinctions are critical and demonstrate the novelty of our approach in leveraging cross-attention for V-ICL in generative diffusion models.
>
> &nbsp;
>
> ---
>
> ### **Clarification on the contributions of our work as it stands in the research of Visual In-Context Learning**
>
> We wish to contextualize our contributions within the field of V-ICL research. Unlike ICL research in NLP, visual ICL is still in its early stages. While ICL properties emerge naturally in LLMs, to date, no prior work has demonstrated that current vision foundation models are inherently capable of V-ICL.
> Existing attempts, such as Visual Prompting (NeurIPS’22), IMProv, Painter (CVPR’23), and LVM (CVPR’24), enable V-ICL through training on curated or uncurated datasets, undermining the core benefits of ICL: the ability to adapt to novel tasks without further training or model updates, and limited generalization performance (as we highlight in Figure 4 of the main paper and Section E of the supplementary).
>
> In contrast, our work addresses this gap by uncovering V-ICL properties in Stable Diffusion using an entirely training-free approach for the first time, aligning more closely with the original definition of ICL in LLM research, while yielding significantly superior generalization performance (a core benefit in ICL). This represents a significant step forward in V-ICL research and demonstrates the feasibility of leveraging pre-trained generative priors for task adaptation without additional supervision.
>
> &nbsp;
>
> ---
>
> [14] Chen et al., "Anifacediff: High-fidelity face reenactment via facial parametric conditioned diffusion models." [arXiv'24]
>
> [15] Karras et al., "Dreampose: Fashion image-to-video synthesis via stable diffusion." [ICCV'23]
>
> [16] Tian et al., "Diffuse, attend and segment: Unsupervised zero-shot segmentation using stable diffusion." [CVPR'24]
>
> [17] Kotovenko et al., "Cross-Image-Attention for Conditional Embeddings in Deep Metric Learning", [CVPR'23]
>
> [18] Lin et al., "Few Shot Medical Image Segmentation with Cross Attention Transformer", [MICCAI'23]
>
> [19] Hossain et al. "Visual Prompting for Generalized Few-shot Segmentation: A Multi-scale Approach", [CVPR'24]
>
> [20] Lu et al., "ViLBERT: Pretraining task-agnostic visiolinguistic representations for vision-and-language tasks" [NeurIPS'19]
>
> [21] Rahman et al., "TriBERT: Human-centric audio-visual representation learning" [NeurIPS'21]

---

> ### Comment · Reviewer_F13E · 2024-12-03
>
> Thanks for author’s feedback. From your feedback, I still have concerns that 1) the 1-shot performance is not competitive to other V-ICL models. 2) The ensemble mechanism you proposed is actually designed for addressing the uniform weight during the feature ensemble. In other words, your implementation is could be used for SegGPT since you can directly use your proposed weights for its mean calculation (you can split those composite images in the feature space, as have done in SegGPT. I understand it would be difficult to implement such a method in MAE-VQGAN since images are combined and resized into 1 image. But for SegGPT and Painter, they just concatenate two images at the H dimension. So technically speaking, they are not composited as same as MAE-VQGAN. Making the use of internal representation to construct qkv weights should not be restricted as you mentioned.  Even as you claim, such an ensembling mechanism would face a limited application. 3) ICL is also a special form of few-shot learning. So the rebuttal from the aspect of other field like Few-shot Learning is weird. 4) No experimental results of intuitive VAE-oriented task vector baseline is provided, so the advantage of using iterative denoising is unknown, let alone such an iterative process would bring extra costs. 5) In my opinion, no use of text is a weaken exploration on SD.
>
> Overall. Based on the all responses from authors, I feel this work is still lacking some important exploration in using SD for ICL, which could be refined for next-round submission, thus I vote for rejection.

---

### Official Review · Reviewer_HFYR · 2024-10-30

**Soundness:** 3
**Presentation:** 3
**Contribution:** 3
**Rating:** 6
**Confidence:** 3

**Summary:**

This paper explores the adaptation of off-the-shelf Stable Diffusion models for visual in-context learning (V-ICL), demonstrating their capability to perform various visual tasks without explicit fine-tuning. By implementing in-place attention re-computation within the self-attention layers of the Stable Diffusion architecture, the model incorporates context between the query and example prompts. This approach effectively adapts the model to six out-of-domain tasks, including foreground segmentation and semantic segmentation, achieving significant performance improvements, such as an 8.9% increase in mIoU for foreground segmentation on the Pascal-5i dataset compared to recent methods. Inspired by feature ensembling in SegGPT, this method also leverages multiple prompts through implicitly-weighted prompt ensembling, enhancing performance across all tasks.

**Strengths:**

- **Originality:** The proposed traning-free visual in-context learning method is is highly innovative. It proposes a re-purpose technique on the self-attention layers of SD. It integrates attention map contrasting, swap-guidance, and AdaIn mechanisms to enhance prediction quality. While these techniques are inspired by existing work, they are effectively incorporated into the overall framework, contributing to the novelty of the approach.
- **Quality:** The primary innovative technique, in-place attention re-computation, is well-motivated and technically ssound. Experimental results validate its effectiveness, demonstrating a solid foundation for the proposed method..
- **Clarity:** The paper is well-written and clearly presented, making it easy for readers to understand the concepts and techniques involved..
- **Significance:** The proposed method significantly outperforms existing approaches on a variety of widely recognized out-of-domain tasks, underscoring the strong impact and benefits of this work.

**Weaknesses:**

- **Comparison methods:** This work compares the proposed method against only two existing approaches, which limits the strength of the comparative analysis. Incorporating additional methods (on different tasks) for comparison would enhance the validity and robustness of the results.

**Questions:**

- SD-VICL involves numerous denoising steps, leading to significantly high inference times. While some studies outside the scope of in-context learning have shown that specific intermediate steps can yield satisfactory results. For instance, references [1] and [2] demonstrate effective outcomes for keypoint detection, while reference [3] focuses on semantic segmentation. Whether the proposed method could leverage a one-step adaptation approach to enhance efficiency without sacrificing performance?

_[1] Tang, L., Jia, M., Wang, Q., Phoo, C.P. and Hariharan, B., 2023. Emergent correspondence from image diffusion. Advances in Neural Information Processing Systems, 36, pp.1363-1389._
_[2] Zhang, J., Herrmann, C., Hur, J., Polania Cabrera, L., Jampani, V., Sun, D. and Yang, M.H., 2024. A tale of two features: Stable diffusion complements dino for zero-shot semantic correspondence. Advances in Neural Information Processing Systems, 36._
_[3] Barsellotti, L., Amoroso, R., Cornia, M., Baraldi, L. and Cucchiara, R., 2024. Training-Free Open-Vocabulary Segmentation with Offline Diffusion-Augmented Prototype Generation. In Proceedings of the IEEE/CVF Conference on Computer Vision and Pattern Recognition (pp. 3689-3698)._

---

> ### Author Response · Authors · 2024-11-25
> **Response to Reviewer HFYR [1/2]**
>
> We appreciate your valuable feedback and suggestions, which have enhanced the quality of the paper. Please find below additional comparisons to existing approaches and our response to the concerns raised regarding inference speed.
>
> &nbsp;
>
> ---
>
> ### **Limited comparisons**
>
> Our intention of restricting the comparisons to IMProv [1] and Visual Prompting [2] models was solely to ensure fair assessments, as they were the only methods that did not rely on task specific datasets such as ours. Nonetheless, we acknowledge the reviewer's concerns and have extended the evaluations to other models like Painter [3] and LVM [4] as shown in Table 4.
>
> - Painter: A ViT-Large model trained on annotated datasets such as COCO, ADE20K, and NYUv2.
>
> - LVM: Built on OpenLLaMA’s 7B-parameter model and trained on the UVD-V1 dataset, which combines 50 datasets (e.g., LAION5B) with both annotated and unannotated images.
>
> As shown in Table 4, our method outperforms both Painter and LVM across multiple tasks.
>
> Additionally, Painter and LVM often suffer from overfitting to training tasks, leading to poor generalization when exposed to novel tasks. For example, when the Painter network was provided with images from the NYUv2 dataset with edge images as prompts, it ends up predicting depth maps instead of edge detection output, indicating the model being overfit to the dataset on which it was trained. See Section E in the revised supplementary for details and visualizations.
>
> Additionally, LVM exhibits inconsistencies in its outputs, indicating that it is unable to infer the task appropriately based on the given input prompts. Specifically, for a given task, despite the example prompts coming from the same task, the model predicts outputs for other tasks in some cases. For example, when given prompts from foreground segmentation, the network sometimes predicts outputs that belong to other tasks such as keypoints, segmentation maps, or RGB images. See Section E in the revised supplementary for details and visualizations.
>
> These models attempt to apply learned task-specific priors instead of inferring the task information from the relationship between the prompt image and its ground truth, as expected by in-context learning.  In contrast, SD-VICL shows superior ability to infer the task and context through the query and prompt inputs, instead of any learned fixed task-specific priors. These results validate our method’s ability to handle novel tasks without overfitting or requiring extensive training on annotated datasets.
>
> We further compare our method against task-specific unsupervised models that build on the Stable Diffusion priors, as described in Tables 1 and 2 in the response to reviewer bhAU's comments.
>
> #### Table 4: Extended comparisons against Painter and LVM
>
> | Model             |    FG Seg. (mIoU ↑)   |   Obj. Det. (mIoU ↑)   |   Edge Det. (MSE ↓ / LPIPS ↓)   |   Color. (LPIPS ↓ / FID ↓)   |
> |--------------------|-----------------------|-------------------------|---------------------------------|-----------------------------|
> |     Painter       |        55.09         |          54.28          |      0.0926 / 0.7294           |      0.3474 / 64.16        |
> |       LVM         |        50.98         |          52.67          |      0.0499 / 0.4259           |      0.3142 / 56.40        |
> | **SD-VICL (Ours)**|     **55.49**        |       **57.10**         |   **0.0213** / **0.1216**      |   **0.2272** / **44.84**   |
>
>
> &nbsp;
>
> ---
>
> ### **Question: Possibility of a one-step adaptation approach**
>
> We appreciate this suggestion and agree that inference time is a challenge for diffusion-based methods. The possibility of one-step adaptation like [5-7] is something we can explore as future work. Separately, exploring faster diffusion strategies such as [8,9] is also a potential avenue for future work to reduce inference time.
>
> In addition, we would like to bring the reviewer's attention to Section 3.2 [L497:527] in the main paper, where we show that using multiple prompts via IWPE is one approach to reduce inference time. For example, with five prompts, performance surpasses the single-prompt while reducing inference time by approximately 1.5×.

---

> ### Author Response · Authors · 2024-11-25
> **Response to Reviewer HFYR [2/2]**
>
> ---
>
> [1] Xu et al. "IMProv: Inpainting-based Multimodal Prompting for Computer Vision Tasks" [arXiv'23]
>
> [2] Bar et al. "Visual prompting via image inpainting" [NeurIPS'22]
>
> [3] Wang et al. "Images speak in images: A generalist painter for in-context visual learning" [CVPR'23]
>
> [4] Bai et al. "LVM: Sequential modeling enables scalable learning for large vision models" [CVPR'24]
>
> [5] Tang et al. "Emergent correspondence from image diffusion" [NeurIPS 2023]
>
> [6] Zhang et al. "A tale of two features: Stable diffusion complements dino for zero-shot semantic correspondence" [NeurIPS'24]
>
> [7] Barsellotti et al. "Training-free open-vocabulary segmentation with offline diffusion-augmented prototype generation"  [CVPR'24]
>
> [8] Yin et al. "Improved distribution matching distillation for fast image synthesis" [NeurIPS'24]
>
> [9] Bolya et al. "Token merging for fast stable diffusion" [CVPR'23]

---

> > ### Comment · Reviewer_HFYR · 2024-11-26
> >
> > Thank the authors for addressing my concerns. Good work! I'd like to keep my positive score.

---

### Official Review · Reviewer_bhAU · 2024-11-03

**Soundness:** 2
**Presentation:** 3
**Contribution:** 2
**Rating:** 5
**Confidence:** 5

**Summary:**

The paper demonstrates that stable diffusion models, trained with vast amount of image data, are adept at in-context learning like the foundational LLMs. The paper heavily draws inspiration from Bar et al. Neurips 2022, where they demonstrated that generative models can be made to perform in-context learning with appropriate training. The proposed model leverages the power of diffusion model, effectively creating a form of cross-attention between query and prompt images and ground truth and proposing an approach to aggregate if multiple in-context prompts are provided.

**Strengths:**

The paper shows an interesting observation, that stable diffusion models, like foundational  LLMs are good at in-context learning, with subtle design choices and engineering. It showed better in-context learning compared to ImProv and VisualPrompting, although in fairness, they were using a weaker VQGAN models, which were not trained over vast data like diffusion. One key advantage of the proposed approach though, is exploiting the emergent properties of Diffusion Models rather than having to train Diffusion models to be able to do In-Context Learning.

**Weaknesses:**

Although the paper highlights an interesting emergent property of the diffusion model, my main concern is the lack of technical contributions. Diffusion models, in my opinion, inherently outperform VQGANs, as they have been trained on vast datasets and have shown excellent performance across various unsupervised tasks, such as keypoint detection (Hedlin et al., CVPR 2024), classification (Li et al., CVPR 2023), and segmentation (Tian et al., CVPR 2024). Additionally, using distinct key-query and value vectors as a form of cross-attention is quite common in models like VILBERT (NeurIPS 2019) and TRIBERT (NeurIPS 2022). Furthermore, we may observe similar in-context emergent properties with multi-modal LLMs like LLAVA or similar models, which highlight the strength of these models being trained on huge datasets. Overall, I disagree with the authors’ claim of a "novel pipeline" as a contribution. While the observation of in-context learning as an emergent property of diffusion is interesting, I believe it reflects the strengths of the diffusion model itself rather than a technical innovation by the authors.

**Questions:**

My major concern lies in the authors' technical contributions, as outlined in the weakness section, and in the fairness of comparing diffusion models with VQGAN-based models, given that diffusion models have already been shown to be superior. I would expect the authors to clarify their contributions more clearly.

My second concern is whether this emergent property is unique to diffusion models or if similar properties are observed in other VLMs or multi-modal LLMs, such as LLAVA.

---

> ### Author Response · Authors · 2024-11-25
> **Response to Reviewer bhAU [1/3]**
>
> Thank you for your valuable feedback. In accordance with your suggestions, we have clarified the contributions, outlined the distinctions with other methodologies, and incorporated additional comparisons to related research. Please find below detailed responses to your valuable feedback.
>
> &nbsp;
>
> ---
>
> ### **Diffusion models, as trained on vast datasets, have shown excellent performance across various unsupervised tasks**
>
> We agree with the reviewer that Stable Diffusion models inherently capture richer priors since they have been trained on vast datasets. As recommended, we compare our approach with task-specific Stable Diffusion based unsupervised methods Tian et al., 2024 and Hedlin et al., 2024 [1,2] and list the following benefits:
>
> - Unlike [1,2] which are task specific, the proposed approach is capable of generalizing to multiple tasks without any task-specific modifications.
>
> - The proposed approach has the ability to benefit from example prompts enabling better overall performance.
>
> - Stable Diffusion-based unsupervised methods primarily utilize the learned feature/attention maps in a post-processing step to predict the task output, disregarding the generative process itself to generate the task outputs. In contrast, our pipeline integrates the generative process itself to generate the task outputs, thereby enhancing the quality of the outputs.
>
> - As presented in Table 1 and Table 2, our proposed method demonstrates superior performance compared to [1] and [2] for Semantic Segmentation and Keypoint Detection, respectively. In the semantic segmentation task, we achieve absolute improvements of 10.3% in mIoU and 6.7% in accuracy, surpassing the results of [1]. Note that we use the same evaluation protocol as [1] (i.e., Hungarian matching) to ensure a fair comparison. Similarly, in the keypoint detection task, we demonstrate improvements of 32.4% in MSE and 12.2% in PCK, outperforming the results of [2].
>
> #### Table 1: Evaluation of semantic segmentation on Cityscapes
>
> | Method             | Eval. Method | mIoU ↑    | Acc. ↑    |
> |--------------------|--------------|-----------|-----------|
> | Tian et al., 2024 [1] | Hungarian    | 21.2      | 76.0      |
> | **SD-VICL (Ours)** | Hungarian    | **31.52** | **82.72** |
>
> #### Table 2: Evaluation of keypoint detection on DeepFashion
>
> | Method             | MSE ↓       | PCK ↑     |
> |--------------------|-------------|-----------|
> | Hedlin et al., 2024 [2] | 6.46      | 70.0      |
> | **SD-VICL (Ours)** | **4.37**   | **82.24** |
>
> Furthermore, we compare our approach with training-based V-ICL approaches such as Painter [3] and LVM [4], as depicted in Table 3. It is noteworthy that Painter is ViT-Large model trained on multiple datasets and LVM is based on the 7B-parameter model of OpenLLama [5] trained on a vast dataset &ndash; UVD-V1 &ndash; which is a unified dataset combining 50 datasets (including 1.5B subset of LAION5B).
> We demonstrate that our method yields superior generalization performance compared to these models across multiple tasks, despite the fact that Painter/LVM require some form of training to enable in-context learning.
>
> #### Table 3: Extended comparisons against Painter and LVM
>
> | Model             |    FG Seg. (mIoU ↑)   |   Obj. Det. (mIoU ↑)   |   Edge Det. (MSE ↓ / LPIPS ↓)   |   Color. (LPIPS ↓ / FID ↓)   |
> |--------------------|-----------------------|-------------------------|---------------------------------|-----------------------------|
> |     Painter       |        55.09         |          54.28          |      0.0926 / 0.7294           |      0.3474 / 64.16        |
> |       LVM         |        50.98         |          52.67          |      0.0499 / 0.4259           |      0.3142 / 56.40        |
> | **SD-VICL (Ours)**|     **55.49**        |       **57.10**         |   **0.0213** / **0.1216**      |   **0.2272** / **44.84**   |

---

> ### Author Response · Authors · 2024-11-25
> **Response to Reviewer bhAU [2/3]**
>
> ---
>
> ### **Use of distinct query and key-value vectors as a form of cross-attention is quite common in models like ViLBERT and TriBERT**
>
> We acknowledge that employing distinct query, key-value vectors in cross-attention is not a novel concept, as demonstrated in ViLBERT [6] and TriBERT [7].
> Nevertheless, our method differs in several key aspects:
>
> - **Use for in-context learning**: To the best of our knowledge, this specific form of cross-attention which models task and context relationships, is being utilized for the first time, particularly to facilitate in-context learning.
>
> - **Distinct Q, K, V sources**: While prior cross-attention methods derive the key-value pair from the same source, our formulation uses Q from the query image, K from the prompt image, and V from the prompt groundtruth, which amounts to three distinct images [L244-249], enabling explicit task-context infusion. This design is discussed in L454-476, where alternative attention formulations using the key-value pair from the same source such as {$ Q_C, K_B, V_B $} and {$Q_D, K_B, V_B$} yield inferior performance compared to our formulation.
>
> - **Training-free**: Unlike ViLBERT/TriBERT, which fine-tune their model once the cross-attention formulation is in effect, our approach does not require any training.
>
> To address any potential ambiguity, we will include explicit references to these prior works in the revised submission and expand our discussion in the corresponding sections to clarify how our approach uniquely applies this concept for in-context learning.
>
> &nbsp;
>
> ---
>
> ### **We may observe similar in-context emergent properties with multi-modal LLMs**
>
> We agree with the reviewer that in-context emergent properties are likely associated with large models being trained on large datasets. In this work, we set out to enable the use of these properties of foundational models for training-free visual in-context learning, which is a challenging task. Please note that this possibility has not been previously investigated in prior research.
>
> &nbsp;
>
> ---
>
> ### **Lack of technical contributions and novelty**
>
> Thank you for the feedback. We believe the contributions can be better re-phrased as follows:
>
> In contrast to LLMs, due to the diverse nature of vision tasks and their outputs, it is more challenging to directly utilize existing vision foundation models for visual in-context learning. To address this challenge, all previous V-ICL methods require some form of training that deviates from the in-context learning paradigm prevalent in LLM works. In contrast, our method introduces the **first training-free V-ICL framework** using the Stable Diffusion model.
>
> While we agree that our approach harnesses the learned generative priors of Stable Diffusion, it is important to note that **uncovering V-ICL properties** remains a **challenging** endeavor, and the Stable Diffusion model cannot be directly employed for in-context learning of diverse downstream tasks. To overcome this while enabling a training-free approach, we introduce modeling of task and context relationships through **in-place attention recomputation**. To the best of our knowledge, we are the first to explicitly enforce context and task inference. Unlike multi-modal prompting methods such as IMProv [8], we achieve this without language guidance, solely through self-attention modifications.
>
> Additionally, as discussed in Section 2.3, we propose **implicitly weighted prompt ensembling (IWPE)** which dynamically weighs prompts based on their query correspondence. This is in contrast to SegGPT's feature ensembling (FE) [9] that uniformly weighs all prompts implying equal informativeness &ndash; which is suboptimal. The proposed IWPE yields absolute improvements of 4.0% in mIoU and 2.9% in accuracy over FE.
>
> &nbsp;
>
> ---
>
> ### **Question: Is the emergent property unique to diffusion models?**
>
> As demonstrated previously in the NLP domain, in-context learning properties are inherent to foundation models trained on extensive datasets. The primary objective of this work was to demonstrate that visual in-context learning using foundation models is feasible under the same stringent definitions and requirements employed in the NLP domain. While achieving this goal, we presented algorithms that incorporate pre-trained diffusion models. However, we anticipate that similar approaches can be extended to other foundation models.

---

> > ### Comment · Reviewer_bhAU · 2024-11-26
> > **Usage of distinct key-value pair from cross-attention.**
> >
> > Although this form of using distinct key-value pairs has not been explicitly explored for in-context learning, the concept of cross-attention between query and support images, to the best of my knowledge, has been studied in works such as [1] Kotovenko et al., CVPR 2023; [2] Lin, Yi, et al., MICCAI 2023; and [3] Hossain et al., CVPR 2023. Moreover, as originally mentioned, TRIBERT [4] employs query, key, and value elements from different modalities. Therefore, I believe the authors' approach is not entirely unique in leveraging cross-attention across multiple sources.
> >
> > That said, I do agree with the authors on one point: the works I mentioned require some form of fine-tuning (with few-shot examples), whereas the authors' approach is training-free.
> >
> > [1] Kotovenko et al., "Cross-Image-Attention for Conditional Embeddings in Deep Metric Learning", CVPR 2023.
> > [2] Lin et al. "Few Shot Medical Image Segmentation with Cross Attention Transformer", MICCAI 2023.
> > [3] Hossain et al. "Visual Prompting for Generalized Few-shot Segmentation: A Multi-scale Approach", CVPR 2024.
> > [4] Rahman et al. "TriBERT: Human-centric audio-visual representation learning", NEURIPS 2021.

---

> > ### Comment · Reviewer_bhAU · 2024-11-26
> > **Overall Comment to the rebuttal**
> >
> > Nonetheless, I believe the main contribution of the work lies in the identification that in-context learning can be effective for diffusion models without the need for explicit training. The authors achieved this through a set of interesting yet simple design choices, although the overall technical contribution is largely derived from previous work and can therefore be considered incremental. Additionally, as mentioned in an earlier comment, there are approaches that enforce cross-attention between task inference and context inference. However, I would be happy to increase my score if the authors address the concerns raised in the previous two comments.

---

> > > ### Author Response · Authors · 2024-11-26
> > > **Response to Reviewer bhAU: "Overall Comment to the rebuttal"**
> > >
> > > We sincerely appreciate the reviewer's feedback and their willingness to revise the scores. We agree with the reviewer that the cross attention has been explored earlier for other works but not in the realm of visual in-context learning. As stated by the reviewer, the significance of this work lies in its ability to showcase how the Stable Diffusion model can be repurposed for visual in-context learning in a training-free manner. We hope that we have addressed the concerns/comments regarding the number of prompts and the cross-attention formulation. We are more than happy to answer any more follow-up questions.

---

> > > > ### Comment · Reviewer_bhAU · 2024-12-01
> > > >
> > > > I feel slightly more positive about the paper than before, thanks to the authors' more elaborate explanations. However, I still believe the paper is somewhat lacking in terms of technical contribution. Additionally, I share the concerns raised by reviewer F13E in his final comments.

---

> > > > > ### Author Response · Authors · 2024-12-02
> > > > > **Response to Reviewer bhAU**
> > > > >
> > > > > We thank the reviewer for their feedback and for feeling more positive about our paper after our previous explanations.
> > > > >
> > > > > Regarding the shared concerns raised by Reviewer F13E, we have addressed these in detail in our response to Reviewer F13E, which can be found at https://openreview.net/forum?id=fKrFTGnoXY&noteId=FN4AL5f4Jt .
> > > > >
> > > > > &nbsp;
> > > > >
> > > > > ---
> > > > >
> > > > > ### **Clarification on the contributions of our work as it stands in the research of Visual In-Context Learning**
> > > > >
> > > > > In addition to the previous response elaborating on our contributions (https://openreview.net/forum?id=fKrFTGnoXY&noteId=FoxGQ4LQZf), to provide additional clarity, we wish to contextualize our contributions within the field of V-ICL research. Unlike ICL research in NLP, visual ICL is still in its early stages. While ICL properties emerge naturally in LLMs, to date, no prior work has demonstrated that current vision foundation models are inherently capable of V-ICL.
> > > > > Existing attempts, such as Visual Prompting (NeurIPS’22), IMProv, Painter (CVPR’23), and LVM (CVPR’24), enable V-ICL through training on curated or uncurated datasets, undermining the core benefits of ICL: the ability to adapt to novel tasks without further training or model updates, and limited generalization performance (as we highlight in Figure 4 of the main paper and Section E of the supplementary).
> > > > >
> > > > > In contrast, our work addresses this gap by uncovering V-ICL properties in Stable Diffusion using an entirely training-free approach for the first time, aligning more closely with the original definition of ICL in LLM research, while yielding significantly superior generalization performance (a core benefit in ICL). This represents a significant step forward in **visual-ICL** research and demonstrates the feasibility of leveraging pre-trained generative priors for task adaptation without additional supervision.

---

> ### Author Response · Authors · 2024-11-25
> **Response to Reviewer bhAU [3/3]**
>
> ---
>
> [1] Tian et al., "Diffuse attend and segment: Unsupervised zero-shot segmentation using stable diffusion" [CVPR'24]
>
> [2] Hedlin et al., "Unsupervised keypoints from pretrained diffusion models" [CVPR'24]
>
> [3] Wang et al., "Images speak in images: A generalist painter for in-context visual learning" [CVPR'23]
>
> [4] Bai et al., "LVM: Sequential modeling enables scalable learning for large vision models" [CVPR'24]
>
> [5] Geng & Liu, "Openllama: An open reproduction of llama, May 2023. URL https://github.com/openlm-research/open_llama"
>
> [6] Lu et al., "ViLBERT: Pretraining task-agnostic visiolinguistic representations for vision-and-language tasks" [NeurIPS'19]
>
> [7] Rahman et al., "TriBERT: Human-centric audio-visual representation learning" [NeurIPS'21]
>
> [8] Xu et al., "IMProv: Inpainting-based Multimodal Prompting for Computer Vision Tasks" [arXiv'23]
>
> [9] Wang et al., "SegGPT: Segmenting everything in context" [ICCV'23]

---

> ### Comment · Reviewer_bhAU · 2024-11-26
> **Clarification about the experiments**
>
> I appreciate the effort of the authors for their detailed rebuttal. While the results presented are intriguing, particularly in demonstrating the generalization capabilities of the proposed training-free approach, I would like some clarification: how many example prompts were provided for each task? This is important for better understanding on my part and ensuring that the comparison is fair. Expecting further clarification from the authors.

---

> > ### Author Response · Authors · 2024-11-26
> > **Response to Reviewer bhAU: "Clarification about the experiments"**
> >
> > We apologize for not being sufficiently clear. For all experiments reported in the rebuttal earlier, we use five prompts to fully demonstrate the combined potential of our training-free V-ICL framework and the IWPE prompt ensembling method. We refer the reviewer to Tables 1 and 2 of the main paper, where we report metrics for both single and five prompts for all tasks. Additionally, the ablation study on the number of prompts [L497-527 in the main paper] provides further insights into the impact of using multiple prompts.
> >
> > For completeness, we have updated Tables 1 and 2 in the rebuttal to explicitly present results for both single-prompt and five-prompt cases. As evident from updated tables, while our approach benefits from prompt ensembling, our single-prompt results demonstrate competitive performance, surpassing the results obtained by related works on both semantic segmentation and keypoint detection tasks.
> >
> > #### Table 1: Evaluation of semantic segmentation on Cityscapes
> >
> > | Method             | Eval. Method | mIoU ↑    | Acc. ↑    |
> > |--------------------|--------------|-----------|-----------|
> > | Tian et al., 2024 [1] | Hungarian    |  $21.2$      | $76.0$      |
> > | Ours - 1 prompt | Hungarian    | $\underline{28.28}$ | $\underline{76.34}$ |
> > | Ours - 5 prompts | Hungarian    | $\textbf{31.52}$ | $\textbf{82.72}$ |
> >
> > #### Table 2: Evaluation of keypoint detection on DeepFashion
> >
> > | Method             | MSE ↓       | PCK ↑     |
> > |--------------------|-------------|-----------|
> > | Hedlin et al., 2024 [2] | $6.46$      | $70.0$      |
> > | Ours - 1 prompt | $\underline{5.36}$   | $\underline{77.19}$ |
> > | Ours - 5 prompts | $\textbf{4.37}$   | $\textbf{82.24}$ |
> >
> > In Table 3 of the rebuttal where we compare our approach with LVM and Painter, five prompts are used for SD-VICL and LVM.
> >
> > Painter employs a grid-based input mechanism where prompts and the query image are stitched together into a composite image. Hence, increasing the number of prompts in a grid-like setup reduces the effective resolution of each image in the grid, leading to degraded performance. This limitation is discussed in Section 2.3 of the main paper and has also been noted by Xu et al. (2023) [8] - Table 8. Consequently, for Painter, we report results for the single-prompt case to reflect their best performance.

---

> ### Author Response · Authors · 2024-11-26
> **Response to Reviewer bhAU: "Usage of distinct key-value pair from cross-attention"**
>
> We appreciate the reviewer's insights regarding prior works on the use of cross-attention, which we will discuss in the revised related work section. We agree with the reviewer on the concept of cross-attention being used in past literature for diverse tasks such as metric learning [10], few-shot segmentation [11,12], and multi-modal representation learning [6,7], where they formulate relationships between different inputs. As the reviewer has duly noted, these methods often require to be fine-tuned to effectively capture the correspondences.
>
> While it is true that TriBERT utilizes three modalities in computing its cross-attention, we would like to further clarify our point regarding the use of *distinct* $Q$, $K$, and $V$ values in our cross-attention formulation.
>
> TriBERT's cross-attention mechanism, as described in its paper and code, is formulated as:
> \begin{equation}
>     \qquad \qquad f = \text{softmax} \left ( \frac{Q_1 (K_2 \oplus K_3)^T}{\sqrt{d}} \right ) (V_2 \oplus V_3 ) \tag{1}
> \end{equation}
> where the subscripts of $Q$, $K$, and $V$ indicate the three modalities ($1,2,3$), and $\oplus$ denotes the concatenation operation in the feature dimension.
>
> In contrast, our cross-attention formulation, as given in equation (7) of the main paper, is as follows:
> \begin{equation}
>     \qquad \qquad f = \text{softmax} \left ( \frac{Q_C K_A ^T}{\tau \cdot \sqrt{d}} \right ) V_B \tag{2}
> \end{equation}
> where  $A$, $B$, and $C$ represent three distinct images: the prompt image, the prompt groundtruth, and the query image, respectively.
>
>
> Expanding TriBERT's attention mechanism (equation (1)) reveals the following key properties:
>
> > - **Attention weight computation:**
> The concatenated keys $K_2 \oplus K_3$ are used to compute the attention scores:
> \begin{equation}
>        \qquad \qquad  [ \alpha_2 ; \\ \alpha_3 ]  =
>         \text{softmax} \left ( \frac{Q_1 (K_2 \oplus K_3)^T}{\sqrt{d}} \right )
>     \end{equation}
>     Here, $\alpha_2$ and $\alpha_3$ represent the attention scores between the query $Q_1$ and the keys $K_2$ and $K_3$, respectively. These scores are normalized jointly via softmax across the feature dimension of all keys in $K_2 \oplus K_3$.
>
> > - **Value aggregation:**
> The computed attention weights are applied to the concatenated values of $V_2 \oplus V_3$:
> \begin{equation}
> \qquad \qquad f = \Sigma_i \alpha_{2,i} V_{2,i} + \Sigma_j \alpha_{3,j} V_{3,j} \tag{4}
> \end{equation}
> where $\alpha_{2,i}$ and $\alpha_{3,j}$ are normalized attention weights for $K_2$ and $K_3$, respectively.
>
> > - **Key-Value pairing:**
> While the softmax enables some interaction between the keys $K_2$ and $K_3$ when normalizing the attention weights, the key-value pairing in aggregation remains modality-specific. There is no *direct* interaction between $K_2$ and $V_3$, or between $K_3$ and $V_2$.
>
> This design implies that while TriBERT's attention spans multiple modalities, its *direct* key-value interactions (value aggregations - eq (4)  ) are restricted to within the same modality.
> In contrast, our cross-attention mechanism explicitly links distinct inputs (i.e., $Q_C$ attend to $K_A$ and aggregates $V_B$) enabling direct interactions across three distinct images.
> This explicit interaction across different sources is a defining characteristic of our formulation and facilitates task-context modeling within the visual domain.
>
> In summary, the main differences between our method and TriBERT are: (1) TriBERT’s key and value vectors directly interact only within the same modality, while our formulation explicitly uses key and value vectors from distinct images to model task and context; (2) our approach is entirely training-free, leveraging pre-trained priors, whereas TriBERT requires fine-tuning; and (3) our cross-attention formulation is specifically designed for visual in-context learning, enabling task and context inference in a novel, training-free paradigm.
>
> We hope this clarification highlights the differences between our method and prior works like TriBERT. We will ensure these distinctions are explicitly discussed in the revised manuscript to avoid ambiguity.
>
> &nbsp;
>
> ---
>
> [10] Kotovenko et al., "Cross-Image-Attention for Conditional Embeddings in Deep Metric Learning", [CVPR'23]
>
> [11] Lin et al., "Few Shot Medical Image Segmentation with Cross Attention Transformer", [MICCAI'23]
>
> [12] Hossain et al. "Visual Prompting for Generalized Few-shot Segmentation: A Multi-scale Approach", [CVPR'24]

---

### Meta-Review · Area_Chair_X5Dh · 2024-12-19

**Metareview:**

(a) The paper demonstrates that Stable Diffusion models, adapted for visual in-context learning (V-ICL), can perform various tasks without fine-tuning by utilizing cross-attention between query and prompt images, achieving significant performance improvements across multiple out-of-domain tasks.

(b) Strengths: The paper presents an innovative training-free approach for visual in-context learning using Stable Diffusion models, leveraging their emergent properties instead of requiring specific training. The method integrates attention map contrasting, swap-guidance, and AdaIn mechanisms, enhancing prediction quality through subtle design choices. Experimental results demonstrate the effectiveness of the in-place attention re-computation technique, achieving significant improvements over existing methods. The paper is well-written and clearly presented, making it accessible to readers. Overall, the proposed approach significantly outperforms previous methods across various out-of-domain tasks, showcasing its potential impact.

(c) Weaknesses: The paper's comparison with only a few baseline methods limits the strength of the analysis, and additional methods such as SegGPT, Painter, and LVM should be included for a more comprehensive evaluation. The generalization of the proposed method is also a concern, as the experiments focus mainly on discriminative tasks, and expanding to more diverse tasks like low-light enhancement or in-painting would strengthen the findings. The focus on Stable Diffusion, a multi-modal model, overlooks the potential contributions of more advanced multi-modal models, such as EMU, that could integrate text and vision for better in-context learning. While the paper presents an interesting observation about in-context learning with diffusion models, the lack of novel technical contributions weakens its impact, as diffusion models already exhibit such emergent properties.

(d) The most important reasons for reject are: the paper raises concerns regarding the competitive performance of its 1-shot approach compared to other V-ICL models. The proposed ensemble mechanism seems to be a reimplementation of methods already used in models like SegGPT, and the application of this technique is overly restrictive. Additionally, the paper does not explore the use of text in Stable Diffusion, which weakens its contribution. Overall, the paper requires more exploration and refinement for future submission.

**Additional Comments On Reviewer Discussion:**

(a) Reviewer bhAU expresses concerns that the paper lacks technical contributions, as diffusion models naturally outperform VQGANs due to being trained on larger datasets. The use of cross-attention with distinct key-query and value vectors is already common in models like VILBERT and TRIBERT. The reviewer also notes that similar in-context emergent properties are seen in other large multi-modal models, questioning the novelty of the proposed pipeline. The authors have addressed most of the concerns. The reviewer further points out that the main contribution of the work is identifying that in-context learning can be effective for diffusion models without explicit training, though the technical contribution is incremental and builds on previous work. The Reviewer bhAU shows similar concerns with the Reviewer F13E.

(b) Reviewer HFYR mentions that the paper compares the proposed method against only two approaches, limiting the strength of its comparative analysis. The SD-VICL method requires numerous denoising steps, resulting in high inference times, and it remains unclear if a one-step adaptation could maintain performance while improving efficiency. Additionally, prior studies have shown that intermediate steps can achieve effective results for tasks like keypoint detection and semantic segmentation, raising questions about alternative strategies for optimization. The authors have addressed the issues and the reviewer keeps the score.

(c) Reviewer F13E shows that the paper lacks sufficient baseline comparisons, excluding key methods like SegGPT, Painter, and LVM, and does not evaluate performance without retrieval processes. The study focuses narrowly on discriminative tasks, limiting its generalization, and misses opportunities to explore broader or task-specific challenges like low-light enhancement or in-painting. Additionally, the approach underutilizes the multi-modal potential of stable diffusion, neglecting the integration of text and more advanced multi-modal models like EMU. The initial rebuttal addresses part of the concerns. The reviewer still has more concerns regarding the experimental parts. The authors attempt to make further clarifications, however, they fail to convince the reviewers. They believe that this work is still lacking some important exploration in using SD for ICL, which could be refined for next-round submission.

---

### Decision · Program_Chairs · 2025-01-22

Reject